# SIRT6 is a DNA double-strand break sensor

Lior Onn[1,2], Miguel Portillo[1,2], Stefan Ilic[3], Gal Cleitman[1,2], Daniel Stein[1,2], Shai Kaluski[1,2], Ido Shirat[1,2], Zeev Slobodnik[1,2], Monica Einav[1,2], Fabian Erdel[4,5], Barak Akabayov[3], Debra Toiber[1,2]*

[1]Department of Life Sciences, Ben-Gurion University of the Negev, Beer Sheva, Israel; [2]The Zlotowski Center for Neuroscience, Ben-Gurion University of the Negev, Beer-Sheva, Israel; [3]Department of Chemistry, Ben-Gurion University of the Negev, Beer-Sheva, Israel; [4]Division of Chromatin Networks, German Cancer Research Center (DKFZ), BioQuant, Heidelberg, Germany; [5]Centre de Biologie Intégrative, CNRS UPS, Toulouse, France

**Abstract** DNA double-strand breaks (DSB) are the most deleterious type of DNA damage. In this work, we show that SIRT6 directly recognizes DNA damage through a tunnel-like structure that has high affinity for DSB. SIRT6 relocates to sites of damage independently of signaling and known sensors. It activates downstream signaling for DSB repair by triggering ATM recruitment, H2AX phosphorylation and the recruitment of proteins of the homologous recombination and non-homologous end joining pathways. Our findings indicate that SIRT6 plays a previously uncharacterized role as a DNA damage sensor, a critical factor in initiating the DNA damage response (DDR). Moreover, other Sirtuins share some DSB-binding capacity and DDR activation. SIRT6 activates the DDR before the repair pathway is chosen, and prevents genomic instability. Our findings place SIRT6 as a sensor of DSB, and pave the road to dissecting the contributions of distinct DSB sensors in downstream signaling.

*For correspondence:
toiber@bgu.ac.il

Competing interests: The authors declare that no competing interests exist.

## Introduction

DNA safekeeping is one of the most important functions of the cell, allowing both the transfer of unchanged genetic material to the next generation and proper cellular functioning. Therefore, cells have evolved a sophisticated array of mechanisms to counteract daily endogenous and environmental assaults on the genome. These mechanisms rely on the recognition of the damaged DNA and its subsequent signaling. This signaling cascade triggers responses such as checkpoint activation and energy expenditure, and initiates the DNA repair process (*Bartek and Lukas, 2007*; *Bartek and Lukas, 2003*; *Ciccia and Elledge, 2010*; *San Filippo et al., 2008*; *Hoeijmakers, 2009*; *Iyama and Wilson, 2013*; *Jackson and Bartek, 2009*; *Lieber, 2008*; *Madabhushi et al., 2014*). If DNA damage is not properly recognized, all downstream signaling will be impaired.

Among the various types of DNA damage, the most deleterious are double-strand breaks (DSBs), which can cause translocations and the loss of genomic material. Until now, very few DSB sensors have been identified, among them poly ADP-ribose polymerase-1 (PARP1), the MRN complex (MRE11, RAD50, NBS1) and Ku70/80 complex. All of these sensors initiate downstream signaling cascades which usually lead to the activation of specific repair pathways, such as homologous recombination (HR) or classical non-homologous end joining (C-NHEJ) (*Andres et al., 2015*; *Sung et al., 2014*; *Woods et al., 2015*). How a specific repair pathway is chosen is not fully understood, but it is known that the identity of the DSB sensor influences the outcome. For example, the MRN complex is associated with HR, whereas Ku70/80 is associated with C-NHEJ. Once DNA damage is recognized, transducers from the phosphoinositide 3-kinase family (e.g., ATM, ATR, and DNA-PK) are

**eLife digest** DNA is a double-stranded molecule in which the two strands run in opposite directions, like the lanes on a two-lane road. Also like a road, DNA can be damaged by use and adverse conditions. Double-strand breaks – where both strands of DNA snap at once – are the most dangerous type of DNA damage, so cells have systems in place to rapidly detect and repair this kind of damage.

There are three confirmed sensors for double-strand break in human cells. A fourth protein, known as SIRT6, arrives within five seconds of DNA damage, and was known to make the DNA more accessible so that it can be repaired. However, it was unclear whether SIRT6 could detect the double-strand break itself, or whether it was recruited to the damage by another double-strand break sensor.

To address this issue, Onn et al. blocked the three other sensors in human cells and watched the response to DNA damage. Even when all the other sensors were inactive, SIRT6 still arrived at damaged DNA and activated the DNA damage response. To find out how SIRT6 sensed DNA damage, Onn et al. examined how purified SIRT6 interacts with different kinds of DNA. This revealed that SIRT6 sticks to broken DNA ends, especially if the end of one strand slightly overhangs the other – a common feature of double-strand breaks. A closer look at the structure of the SIRT6 protein revealed that it contains a narrow tube, which fits over the end of one broken DNA strand. When both strands break at once, two SIRT6 molecules cap the broken ends, joining together to form a pair. This pair not only protects the open ends of the DNA from further damage, it also sends signals to initiating repairs. In this way, SIRT6 could be thought of acting like a paramedic who arrives first on the scene of an accident and works to treat the injured while waiting for more specialized help to arrive.

Understanding the SIRT6 sensor could improve knowledge about how cells repair their DNA. SIRT6 arrives before the cell chooses how to fix its broken DNA, so studying it further could reveal how that critical decision happens. This is important for medical research because DNA damage builds up in age-related diseases like cancer and neurodegeneration. In the long term, these findings can help us develop new treatments that target different types of DNA damage sensors.

recruited to the sites of damage. They initiate a broad cascade, recruiting and activating hundreds of proteins which regulate the cellular response, including cell cycle progression, transcription, and metabolism. Ultimately, this response will determine whether the cell will live, senesce, or die. Failure to recognize and repair DSBs may lead to tissue ageing and disease (*Ciccia and Elledge, 2010*; *San Filippo et al., 2008*; *Gasser et al., 2017*; *Ribezzo et al., 2016*; *Shiloh, 2014*).

Sirtuin 6 (SIRT6) is a chromatin-bound protein from a family of NAD$^+$-dependent deacylases and ADP-ribosylases. Through these functions, SIRT6 regulates DNA damage repair (DDR), telomere maintenance, and gene expression (*Feldman et al., 2013*; *Jiang et al., 2013*; *Kugel and Mostoslavsky, 2014*). The importance of SIRT6 to DNA maintenance is exemplified in SIRT6-KO mice phenotypes, which include accelerated ageing, cancer and neurodegeneration (*Kaluski et al., 2017*; *Stein and Toiber, 2017*; *Tasselli et al., 2017*; *Zorrilla-Zubilete et al., 2018*; *Zwaans and Lombard, 2014*). SIRT6-deficient cells exhibit genomic instability, increased aerobic glycolysis and defects in DNA repair, among other phenotypes (*Kugel and Mostoslavsky, 2014*; *Stein and Toiber, 2017*; *Tasselli et al., 2017*). Moreover, it was recently shown that the capacity of SIRT6 to repair DSB, but not to perform nucleotide excision repair (NER), is directly linked to longevity (*Tian et al., 2019*).

We have shown previously that SIRT6 is one of the earliest factors recruited to DSBs, arriving at the damage site within 5 seconds and allowing the opening of chromatin at these sites by recruiting the chromatin remodeler SNF2H (*Toiber et al., 2013*). In addition, the silencing of SIRT6 resulted in impaired downstream signaling, affecting the recruitment of key repair proteins such as Ku80, BRCA1 and 53BP1, among others, which are involved in both NHEJ and HR (*Bunting et al., 2010*; *Chen et al., 2017*; *Daley and Sung, 2014*; *Escribano-Díaz et al., 2013*; *Gupta et al., 2014*; *McCord et al., 2009*; *Tang et al., 2013*; *Toiber et al., 2013*). These studies indicate that SIRT6 plays important roles at very early stages of the DDR. The prominent role of SIRT6 in the early steps of DNA damage signaling raises the fascinating possibility that it is also directly involved in DSB

sensing. In this work, we show that SIRT6 is indeed a DSB sensor, able to detect broken DNA and to activate the DNA damage signaling, revealing its key role in DNA repair initiation.

## Results

### SIRT6 arrives at sites of damage independently of other sensors or signaling

First, we set out to investigate the relationship between SIRT6 and the three known DSB sensors, PARP1, MRE11 (of the MRN complex), and Ku80 (of the Ku complex). PARP proteins are among the fastest known enzymes to arrive at DSBs, and their absence is known to impair the recruitment of DSB repair enzymes such as MRE11, NBS1 and Ku80 (*Haince et al., 2008*; *Yang et al., 2018*). We inhibited PARP activity by supplementing cells with Olaparib, and tracked SIRT6 recruitment to sites of laser induced damage (LID) by live-cell imaging. Interestingly, SIRT6 recruitment was found to be independent of PARP activity. SIRT6 arrived at the damage sites even when PARP proteins were inhibited, while the recruitment of the macro-H2A macro domain, which was used as a control, depended entirely on PARylation (*Figure 1A–C*, *Figure 1—figure supplement 1A–C*).

Subsequently, we silenced MRE11 and observed impaired NBS1 recruitment but no effect on SIRT6 (*Figure 1D–F*, *Figure 1—figure supplement 1D–F*). Ku80 silencing resulted in the expected defects in Ku70 recruitment, but did not impair SIRT6 arrival, in fact even larger amounts of SIRT6 were recruited to the site of damage (*Figure 1D–F*, *Figure 1—figure supplement 1 G-I*). Moreover, when we tested the effect of SIRT6-KO (*Figure 1—figure supplement 1J*) on the recruitment of MRE11 and Ku80, we found that while MRE11 recruitment was defective (*Figure 1G–I*), Ku80 was not affected by the lack of SIRT6 (*Figure 1J–L*). This suggests that SIRT6 may have a role in MRN recruitment or residency at DSB, but that the Ku complex is independent of it. Next, we silenced ATM and H2AX, which are both involved in DDR signaling (*Figure 1—figure supplement 2A*). Even though this produced defective signaling, as shown by decreased DDR signaling (*Figure 1—figure supplement 2B–D*), SIRT6 arrived at the sites of damage independently of these factors (*Figure 1—figure supplement 2E–G*).

These results indicate that SIRT6 recruitment is independent of known DSB sensors and is upstream of ATM and H2AX phosphorylation. To understand whether SIRT6 is recruited through by signaling initiated at the sites of damage themselves, we tested whether it can be recruited by the initiation of a DNA damage response in the absence of actual DNA damage (lack of DSBs). To answer this question we took advantage of a tethering assay in which we used U2OS cells containing 256x lactose operator (LacO) repeats in their genome (*Shanbhag et al., 2010*; *Tang et al., 2013*). We transfected these cells with chimeric proteins containing lactose repressor (LacR) conjugated to known DDR-initiating repair enzymes (scheme in *Figure 2A*; *Soutoglou and Misteli, 2008*). In this system, the mere presence of ATM (ATM-LacR-Cherry) on chromatin initiates the DDR, as shown by H2AX ser-139 phosphorylation (γH2AX) (*Figure 2—figure supplement 1A–B*; *Soutoglou and Misteli, 2008*). However, in this system with no actual DNA damage, ATM failed to recruit SIRT6 to the LacO site, even though signaling was taking place and H2AX was phosphorylated (*Figure 2B–C*). As a control, we showed that known interactors such as SNF2H and Ku80 (*McCord et al., 2009*; *Toiber et al., 2013*) did recruit SIRT6 to the tethering sites (*Figure 2B–C*, *Figure 2—figure supplement 1C–D*). Moreover, MRE11 and NBS1 also recruited SIRT6 to the LacO site (*Figure 2—figure supplement 1C–D*), suggesting that there is either direct interaction between these sensors and SIRT6 or that they work together in a DDR complex.

Taken together, these results indicate that SIRT6 arrives at the sites of damage independently of MRE11, Ku80 and PARP activity, and that signaling itself is not sufficient to bring SIRT6 to the damage sites in the absence of actual DNA damage.

### SIRT6 binds DNA DSBs directly

The findings described so far suggest that SIRT6 responds selectively to the actual damage, and that silencing or inhibiting key factors in the DDR do not affect its fast recruitment. Therefore, we tested whether SIRT6 could detect the actual DNA break on its own. We first measured SIRT6 capacity to bind naked DNA by electrophoretic mobility shift assay (EMSA). We found that SIRT6 was able to bind naked DNA without preference for a sequence (we tested different oligos and restricted

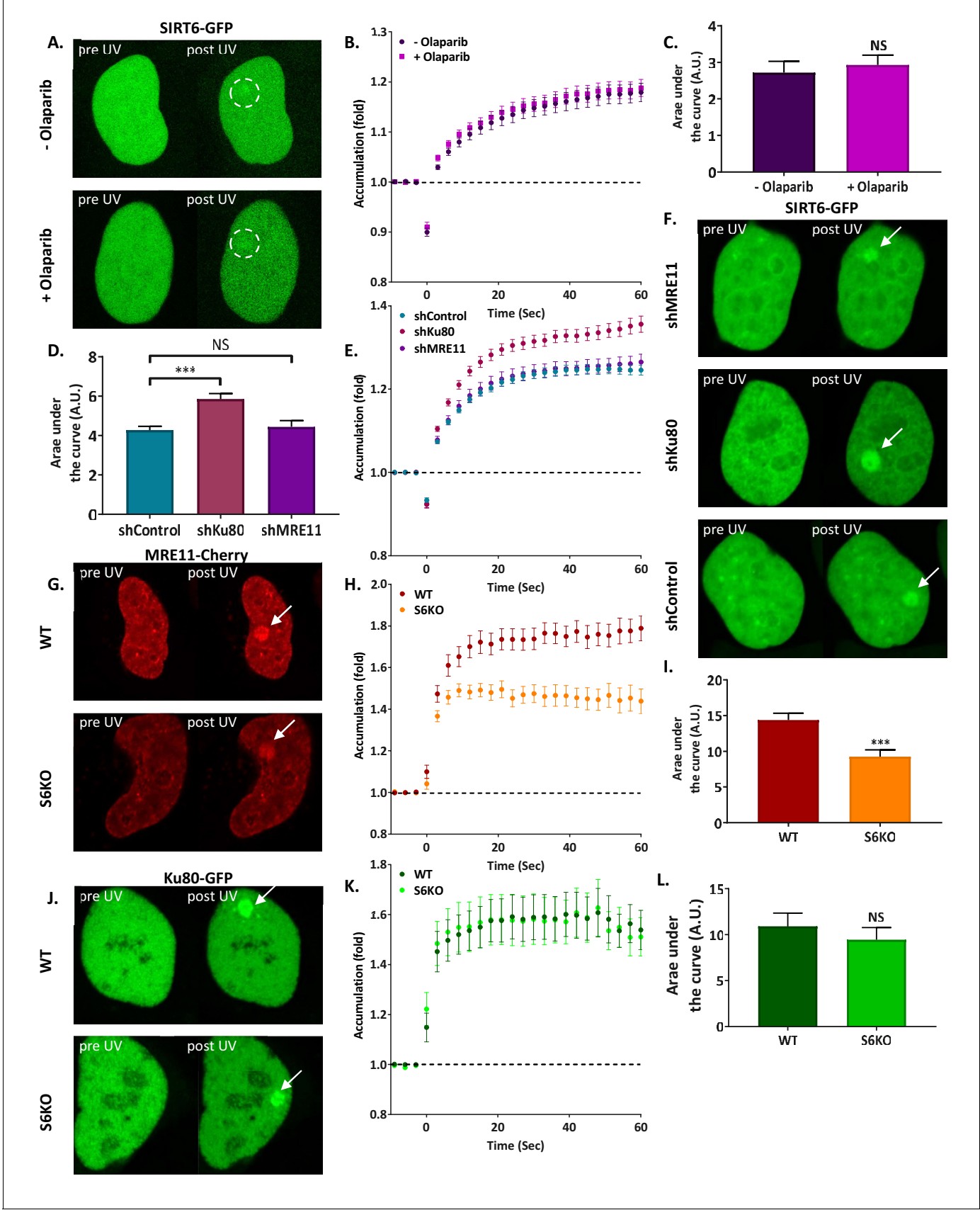

**Figure 1.** SIRT6 arrives at sites of damage independently of other repair factors. (**A–C**) Imaging and AUC for SIRT6-GFP in cells with or without Olaparib. (**A**) Live imaging recruitment upon UV laser-induced damage (LID) shown by SIRT6-GFP in U2OS +/– Olaparib. Representative experiment examining SIRT6 recruitment to LID (n[+Ola]=23, n[–Ola]=23). (**B**) SIRT6 accumulation in same experiment as panel (**A**). (**C**) Average area under the curve (AUC) for cells +/– Olaparib in three replicate experiments. Error bars are the standard error of the mean (SEM) (n[+Ola]=38, n[–Ola]=39, p>0.05). (**D–F**) Imaging and AUC for SIRT6-GFP accumulation in shControl, shKu80 or shMRE11 Hela cells. (**D**) Average AUC from three experiments. Error bars are the SEM (shControl: n = 50; shKu80: n = 50, p<0.0005; shMRE11: n = 52, p>0.05). Accumulation of SIRT6-GFP (**E**) and imaging (**F**) from a representative experiment examining SIRT6 recruitment after LID (n[shControl]=28; n[shKu80]=30, n[shMRE11]=30). (**G–I**) MRE11-Cherry accumulation in response to LID in SIRT6 WT and KO U2OS cells. MRE11-Cherry imaging (**G**) and accumulation (**H**) in a representative experiment (n[WT]=20, n[KO]=16). (**I**) Mean AUC for three replicate experiments. Error bars are the SEM (n[WT]=36, n[KO]=33, p<0.0005). (**J–L**) Ku80-GFP accumulation in response to LID in SIRT6 WT and KO U2OS cells. Ku80-GFP imaging (**J**) and accumulation (**K**) in a representative experiment (n[WT]=17, n[KO]=17). (**L**) Mean AUC for three replicate experiments. Error bars are the SEM (n[WT]=33, n[KO]=33, p>0.05).

The online version of this article includes the following figure supplement(s) for figure 1:

**Figure supplement 1.** SIRT6 arrivesatsites of damage independently of other repair factors.

**Figure supplement 2.** SIRT6 arrivesatsites of damage independently of other repair factors.

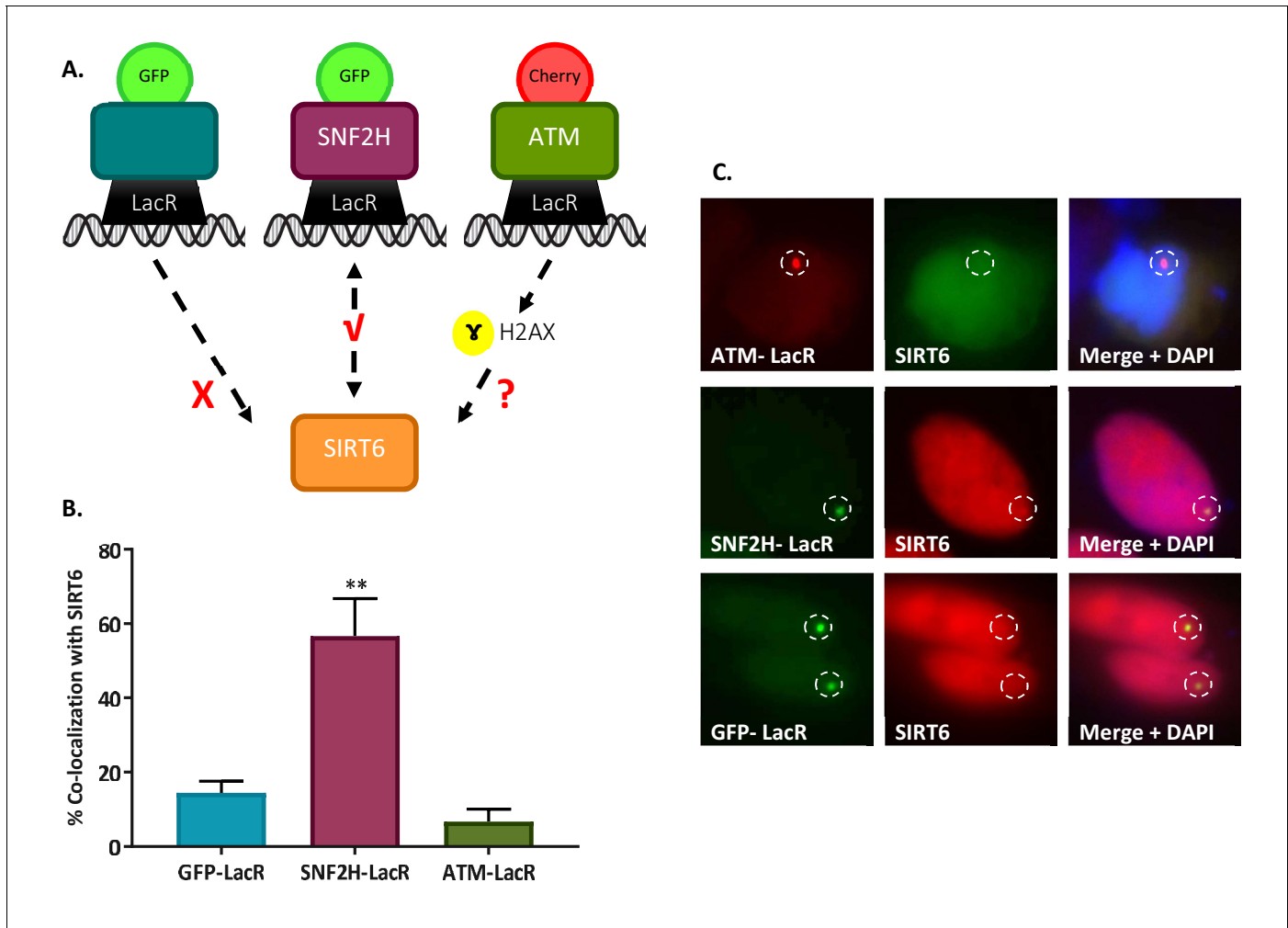

**Figure 2.** SIRT6 is not recruited by signaling. (**A**) Schematic representation of the 'Tethering assay'. Recruitment can occur through DDR signaling (ATM-LacR-Cherry) or through direct protein–protein interaction (SNF2H-LacR-GFP). (**B, C**) Recruitment of SIRT6-GFP/SIRT6-Cherry to LacO sites by ATM-LacR-Cherry (n = 30, p>0.05), SNF2H-LacR-GFP (n = 85, p<0.005) and GFP-LacR (n = 85). The bar chart in panel (**B**) depicts averages for3–6 experiments. Error bars are SEM.

The online version of this article includes the following figure supplement(s) for figure 2:

**Figure supplement 1.** SIRT6 is not recruited by signaling.

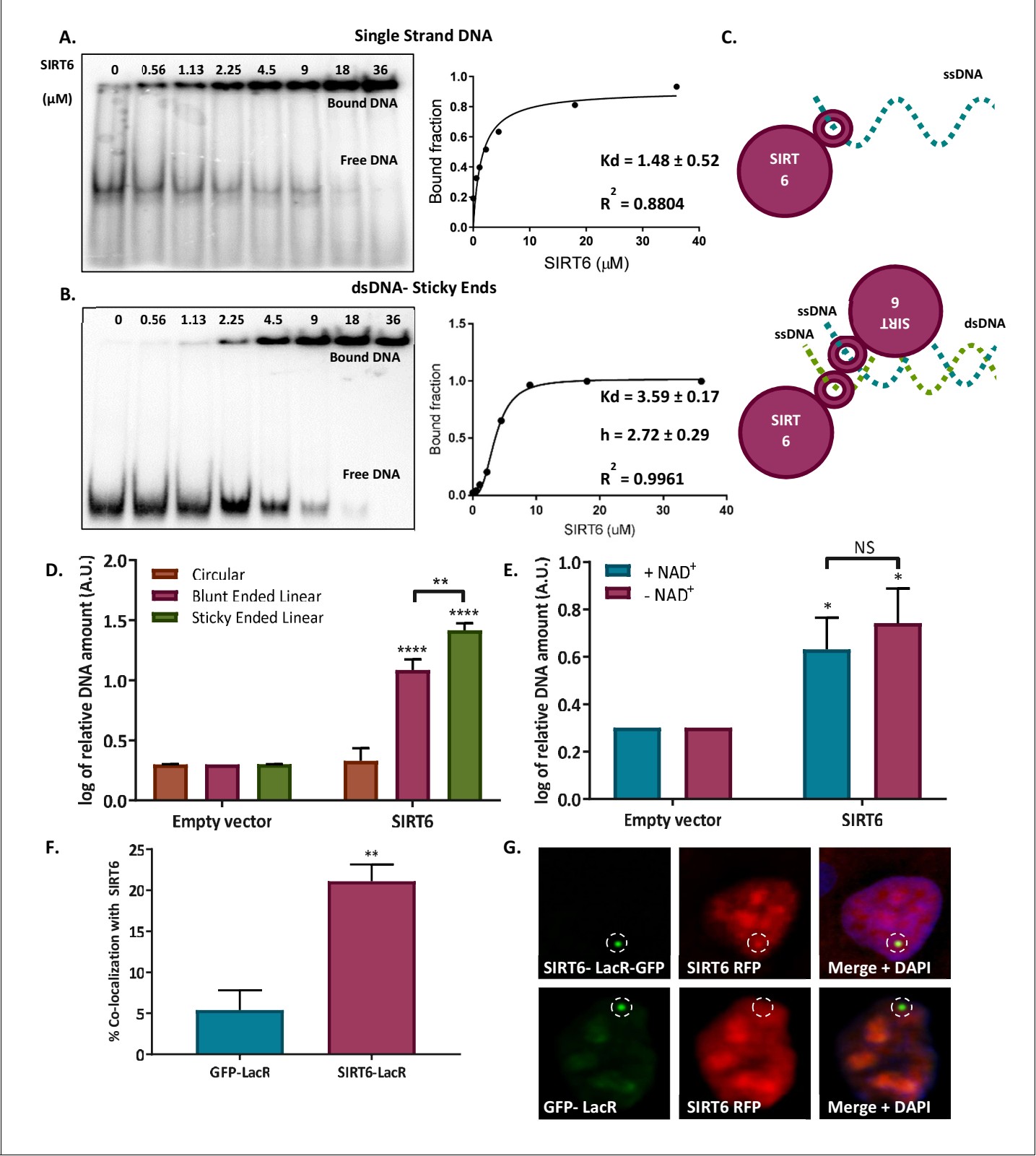

**Figure 3.** SIRT6 binds DNA with no intermediates. (**A–B**) Gel retardation assay of 32 P-5′ end-labeled single-strand DNAs and sticky ended dsDNAs as a function of increasing concentrations of SIRT6-His (ssDNA, Kd = 1.48 ± 0.52; sticky dsDNA, Kd = 3.59 ± 0.17). (**C**) Suggested model of SIRT6 binding to ssDNA as a monomer or open ssDNA ends of dsDNA as a dimer. (**D**) Ability of SIRT6-Flag to bind to the DNA of circular, blunt-ended and sticky-ended cleaved plasmids. The bar chart depicts averages for three replicate experiments (error bars show SEM), after logarithmic transformation. (**E**)
*Figure 3 continued on next page*

*Figure 3 continued*

SIRT6-Flag DNA-binding ability for an open-ended +plasmid +/- NAD. Data are averages from four experiments, with error bars representing SEM (after logarithmic transformation). (F, G) Dimerization of SIRT6 at the LacO site, represented by the recruitment of SIRT6-Cherry by SIRT6-LacR-GFP (n = 181, p<0.005) or GFP-LacR (n = 104). Data are averages from four experiments, with error bars representing SEM.

The online version of this article includes the following figure supplement(s) for figure 3:

**Figure supplement 1.** SIRT6 binds DNA with no intermediates.
**Figure supplement 2.** SIRT6 binds DNA with no intermediates.

---

sites, see *Table 1*) (*Figure 3A–B*, *Figure 3—figure supplement 1A*). We studied the preference of SIRT6 for several DNA damage structures, including dsDNA with blunt or overhanging ends as well as RNA. SIRT6 has the ability to bind them all, but it binds RNA with much lower affinity (*Figure 3— figure supplement 1B*). SIRT6 exhibits the highest affinity towards ssDNA (Kd = 1.39 µM), showing binding affinity values similar to those for MRE11 (Kd ~1 µM) (*Williams et al., 2008*) and Ku80 (Kd = 0.4 µM) (*Arosio et al., 2002*). Interestingly, on the basis of the curve fitting, SIRT6 seems to bind ssDNA at one site as a monomer. By contrast, there seems to be a cooperative effect when testing blunt and sticky-end DNA (Hill Slope greater than 1), suggesting that for open-ended dsDNA, two molecules of SIRT6 participate in binding, each SIRT6 molecule binding one DNA strand (*Figure 3A–B*, *Figure 3—figure supplement 1A*, scheme in *Figure 3C*). As all of the DNAs used in the EMSA were open-ended, we developed an additional DNA-binding assay based on the co-immuno-precipitation of a plasmid (IP-qPCR).

In brief, flag-tagged repair proteins were purified and incubated with DNA, then immunoprecipitated along with the DNA that they bound. The DNA was later purified and its enrichment was measured by qPCR. Proteins were incubated either with a circular plasmid or with the same plasmid presenting blunt or sticky ends. As expected, NBS1, which does not bind DNA by itself (*Myler et al., 2017*), did not bind either plasmid (open or closed ends). By contrast, SIRT6 and MRE11 had high affinity to liner DNA, but they showed almost no binding to closed plasmids (*Figure 3—figure supplement 1C*). Moreover, SIRT6 exhibited a higher affinity for sticky ends, structures that show a high resemblance to DSBs, over blunt ends (*Figure 3D*). In addition, it did not distinguish between 3' or 5' overhangs (*Figure 3—figure supplement 1D*). These assays indicate that SIRT6 does not function by binding intact DNA or a particular sequence, but rather by binding to open DNA ends, and particularly to ssDNA. It is important to note that this capacity is independent of the presence of NAD$^+$, the known cofactor of SIRT6 (*Figure 3E*), and the binding of DNA per se, does not activate SIRT6 catalytic activity (*Figure 3—figure supplement 1E*). Moreover, SIRT6 was able to protect the open ends of DNA from exonuclease activity (ExoI), preventing exonuclease cleavage just as in the case of MRE11 and implying that SIRT6 specifically binds to DNA ends (*Figure 3—figure supplement 1F–H*).

## SIRT6 binds DNA ends as a dimer

Our EMSA results indicate that SIRT6 binds ssDNA with no cooperativity, suggesting a single binding site. By contrast, when the substrates were dsDNA oligos, we found the Hill coefficient to be greater than 1, indicating cooperativity (*Figure 3A–B*, *Figure 3—figure supplement 1A*). These results suggest that a single molecule of SIRT6 binds ssDNA. Even so, given two ssDNAs, such as would be present at an open-ended DSB, one SIRT6 molecule will interact with another, allowing a dimer of SIRT6 to bind a single molecule of dsDNA that has two open ends on a single side, 5' and 3' (see schematic *Figure 3C*). Together, the two SIRT6 molecules show cooperativity.

Interestingly, the known crystal structure of SIRT6 presents a dimer conformation (*Jiang et al., 2013*; *You et al., 2017*). To further characterize the structure of SIRT6 in a solution, we used size exclusion chromatography-multi-angle light scattering (SEC-MALS) and small-angle X-ray scattering (SAXS). Importantly, both methods showed that SIRT6 tends to aggregate; however, when using SEC-MALS, we noted that the aggregation was significantly reduced by the presence of DNA oligomers (*Figure 3—figure supplement 2A*), which suggests that SIRT6 is stabilized by (and favors) DNA interactions. SAXS data provide a low-resolution structure of SIRT6, presumably corresponding to a tetramer (*Figure 3—figure supplement 1B–D*), supporting the model suggested by the EMSA results (with dimers at the 5' and 3', a tetramer). The result obtained by SAXS does not exclude the

presence of SIRT6 dimers or trimers in solution (see *Table 2*). Last, we measured dimerization in vivo by taking advantage of SIRT6-LacR-GFP localization at LacO sites and the recruitment of SIRT6-RFP, observing a significant co-localization of both SIRT6 molecules (*Figure 3F–G*), indicating that the bound SIRT6-GFP recruits the soluble SIRT6-RFP.

Overall, our predictions suggest that the SIRT6-DNA complex is organized in dimers, probably at each end of the DNA oligomers. Moreover, on the basis of the reconstructed SAXS structure, we show a compaction of SIRT6 in the presence of DNA, suggesting a conformational change (*Figure 3—figure supplement 1B–D*).

## SIRT6 binds ssDNA through its core domain, which forms a 'tunnel-like' structure

SIRT6 has not been previously reported in the literature to be a DNA binding protein, so we aimed to identify the domain involved in ssDNA binding. To this end, we first analyzed the SIRT6 structure to find a potential DNA-binding domain. We found a region within the core domain (28 a.a.) that had potential to bind DNA (*Figure 4A–C*). We purified full-length SIRT6 (SIRT6 FL) and a fragment of the core domain alone (core: from a.a. 34 to 274). Both were able to bind DNA with similar affinities, indicating that the core domain is the main domain responsible for DNA binding (*Figure 4D*).

To understand which amino acids could be involved in the DSB binding, we mapped them to the known structure of SIRT6 (http://dnabind.szialab.org/). The model points to a subset of amino acids that are more likely to be involved in DNA binding. Surprisingly, these amino acids are concentrated near a physical structure that resembles a tunnel (*Figure 4A*). This tunnel is narrow and could accommodate ssDNA (*Figure 4E*), but not larger dsDNA. Without an open end, normal undamaged DNA could not enter this tunnel, but broken DNA ends could. Therefore, we hypothesized that the destruction or disruption of the tunnel would impair SIRT6 DNA-binding capacity. To test this hypothesis, we generated several point mutations of the amino acids in the tunnel-like structure of SIRT6 (*Figure 4—figure supplement 1A–B*). Purified SIRT6-MBP point-mutants were tested by EMSA to estimate their DNA-binding ability. As predicted, single point mutations in key amino acids at the tunnel (including the catalytic dead mutant H133Y) impaired the DNA- binding capacity (*Figure 4F–G*). The only mutant that showed no effect on binding was D63Y, in which the mutated amino acid did not impair the charge as strongly as the D63H mutation. Interestingly, mutations in D63 had previously been reported to provoke the loss of SIRT6 function in cancer, and have recently been shown to be lethal in humans (*Ferrer et al., 2018*; *Kugel et al., 2015*).

As our prediction shows that the SIRT6 DNA-binding domain is in close proximity to its catalytic domain, we set out to examine how these mutations would affect SIRT6 catalytic activity. We performed a Fluor-de-lys assay to assess the mutant deacylation activity, using a H3K9-myristolatted peptide. Most mutants showed a decrease in SIRT6 activity compared to SIRT6-WT; however, A13W mutation showed increased SIRT6 activity (*Figure 4—figure supplement 1C*). This finding indicates that DSB binding and SIRT6 deacylation activity are not completely linked. However, given the close proximity of the two domains, they may share some of their functions because of the similarity of ssDNA and NAD$^+$ molecules (ssDNA is a polymer of nucleotides; NAD$^+$ consists of two nucleotides joined through their phosphate groups).

## DNA binding ability is conserved among other Sirtuins

The core domain of SIRT6, where its DNA-binding domain is located, is conserved among all Sirtuins. Therefore, we tested whether other mammalian Sirtuins could bind DSB as well. Our results indicate that all Sirtuins have some capacity to bind broken-ended DNA, but some do it with a significantly lower affinity (*Figure 4H*, *Figure 4—figure supplement 1D*). Only SIRT7 showed binding capacity towards circular DNA, as previously described (*Gil et al., 2013*). It is also important to note that we tested mouse and human SIRT6 (mSIRT-Flag, hSIRT6-His) and found that both bind linear, but not circular DNA (*Figure 4H*, *Figure 4—figure supplement 1D*).

## SIRT6 can initiate DNA damage response

As shown above, SIRT6 directly recognizes DNA breaks and arrives at the sites of damage independently of DDR signaling. Nonetheless, DNA damage recognition per se cannot activate the DDR. Therefore, we set out to examine whether SIRT6 also has the capacity to initiate the DDR through

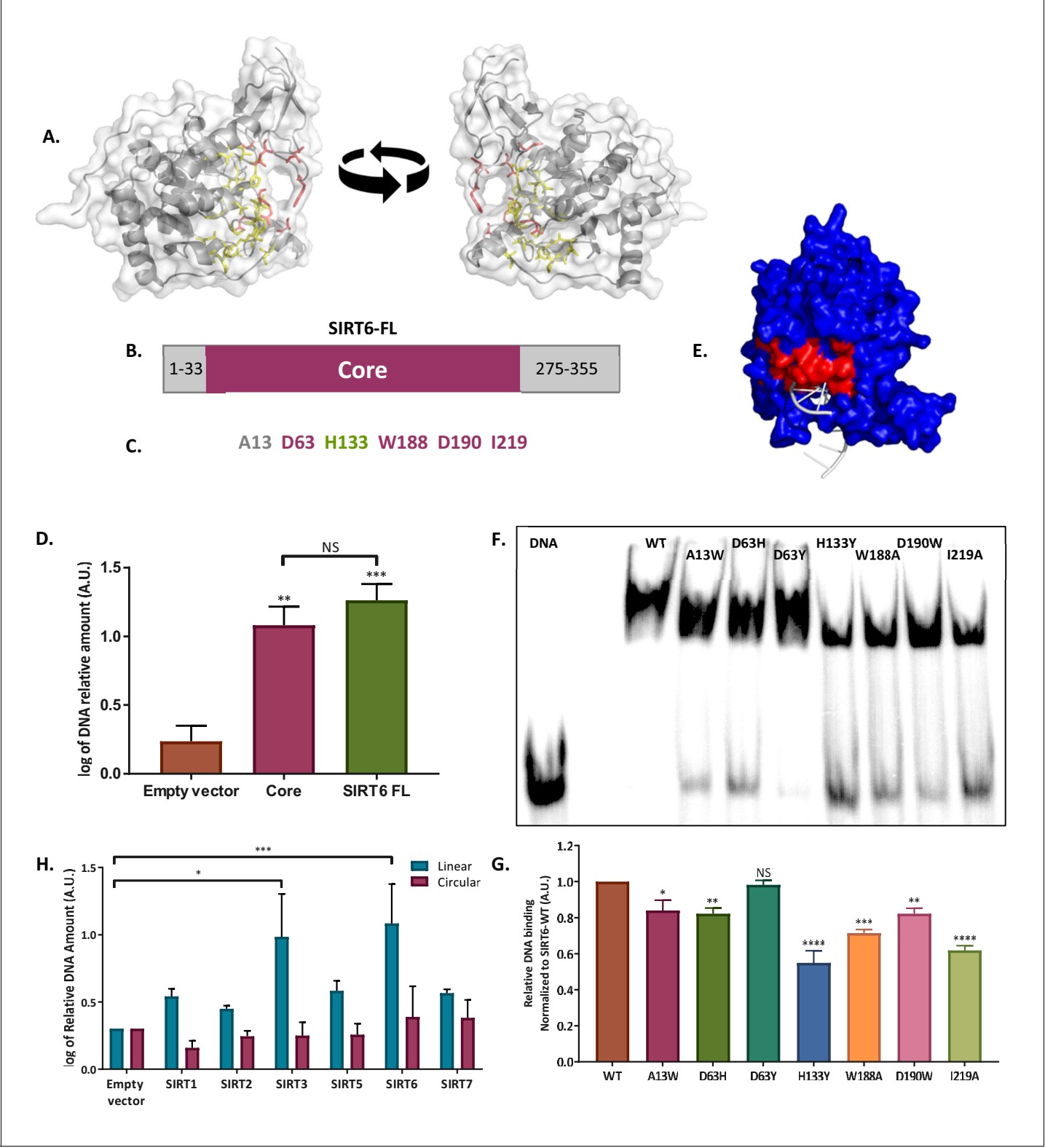

**Figure 4.** SIRT6 binds DSB through its core domain. (**A**) Predicted DNA-binding site based on the published SIRT6 structure, (http://dnabind.szialab.org/). Highlighted in yellow are the predicted DNA-binding amino acids in the SIRT6 core domain; red highlights show the tunnel-forming amino acids that were mutated. (**B**) Schematic representation of the SIRT6 core domain. (**C**) List of amino acids that are predicted to participate in the 'tunnel-like' structure. (**D**) DNA binding of an open-ended plasmid by full-length SIRT6 (p<0.0005) and by the SIRT6-core domain (p<0.005). Data are the log of averages from three experiments (with error bars respresenting SEMs). (**E**) SIRT6 ssDNA-binding prediction, based on the known SIRT6 structure with

*Figure 4 continued on next page*

*Figure 4 continued*

bound ssDNA. (**F, G**) Gel retardation assay of 32 P-5' end-labeled ssDNAs with SIRT6-MBP mutants. Data are averages from three experiments (with error bars representing SEMs). (**H**) Ability of Flag-tagged mammalian Sirtuins to bind the DNA of circular and linear plasmids. Data are averages from 4–7 experiments (with error bars representing SEMs) (*, p <0.05; **, p <0.005; ***, p <0.0005).

The online version of this article includes the following figure supplement(s) for figure 4:

**Figure supplement 1.** SIRT6 binds DSB through itscore domain.

downstream signaling. To that aim, we took advantage of the previously described tethering assay using SIRT6-LacR-GFP/Cherry chimeras. Remarkably, SIRT6 has the same ability to induce the activation of the DDR as MRE11, measured by its capacity, compared to that of LacR-GFP/Cherry, to activate the phosphorylation of H2AX at the LacO site. Interestingly, the SIRT6 catalytic mutant SIRT6-HY was also able to initiate the DDR, raising the possibility that SIRT6 DDR initiation capacity is independent of its catalytic activity (*Figure 5A–B*).

Nevertheless, because SIRT6 can generate dimers, endogenous SIRT6 could dimerize in the cells with SIRT6-HY-LacR, allowing the activation of the DDR. To test this possibility, we used nicotinamide (NAM) to inhibit endogenous SIRT6 activity (*Figure 5—figure supplement 1A*). However, even when the endogenous SIRT6 was inhibited (shown by the increase in H3K56ac), LacR-SIRT6-HY was still able to activate the DDR, supporting the evidence of DDR initiation that is independent of SIRT6 catalytic activity (*Figure 5—figure supplement 1B*).

It is important to highlight that SIRT6-HY has 50% less DNA-binding capacity to DSB than wild-type SIRT6 (*Figure 4F–G*); however, in this assay, it is forced to bind to the DNA through the LacR domain. In fact, we predicted that SIRT6-HY would fail to bind DNA if it was not tethered to chromatin through the LacR domain. To prove this hypothesis, we tested SIRT6-HY recruitment to DSBs in vivo using laser-induced damage in SIRT6-KO U2OS cells. Using SIRT6-KO cells rules out any contribution that an interaction with the endogenous SIRT6 might have. As expected, we found that unlike SIRT6-WT, SIRT6-HY does not arrive at sites of damage (*Figure 5C–E*). This finding strengthens our hypothesis that DNA binding is an important step in the role of SIRT6 in DSB repair, and that the residue that is defective in the SIRT6-HY mutant is critical for SIRT6-DSB binding.

To study whether SIRT6 activity and initiation capacity are separate, we tested Core-LacR-GFP, which has an active catalytic domain but lacks the C and N terminus of SIRT6 (*Tennen et al., 2010*). We observed that Core-LacR-GFP failed to activate the DDR (*Figure 5F*, *Figure 5—figure supplement 1C*), suggesting that other domains play a more prominent role in initiating signaling.

Moreover, we tested the initiation capacity of LacR-SIRT1, SIRT2 and SIRT7 in the tethering assay, because all of these Sirtuins have the ability to localize to the nucleus and have been associated with DNA repair (*Jeong et al., 2007*; *Li et al., 2016*; *Paredes and Chua, 2016*; *Rifai et al., 2018*; *Vazquez et al., 2017*; *Zhang et al., 2016*). Remarkably, SIRT2 and SIRT7 could initiate the DDR, but SIRT1 could not (see note in 'Materials and methods') (*Figure 5G*, *Figure 5—figure supplement 1D*). Although other Sirtuins have some binding activity and some initiation capacity, SIRT6 is unique for having both.

Taken together, these experiments indicate that although SIRT6 binds DNA through its core domain, the activation of downstream signaling does not require the catalytic activity of SIRT6, but its N and C terminus are required for DDR activation.

Last, we tested whether SIRT6 could recruit repair factors of the DDR cascade and whether it shows a preference for a certain repair pathway. Although we observed a more prominent effect of SIRT6 on the recruitment of the HR initiator MRE11 rather than that of the NHEJ initiator Ku80 (*Figure 1G–L*), it was previously reported that SIRT6 affects both repair pathways (*Chen et al., 2017*; *Mao et al., 2011*; *McCord et al., 2009*; *Tian et al., 2019*; *Toiber et al., 2013*). Indeed, we noticed that SIRT6 deficiency results in impaired recruitment of both 53BP1 and BRCA1 to the sites of laser-induced DSBs, suggesting impaired activation of both NHEJ and HR (*Figure 6—figure supplement 1A*).

In order to test SIRT6's ability to recruit these and other DDR factors to the sites of damage, we took advantage of the tethering system once more. Our results show that SIRT6 can recruit proteins that are involved in HR, such as MRE11, NBS1, ATM and BRCA1, as well as proteins that are involved in NHEJ, such as Ku80, Ku70 and 53BP1 (*Figure 6A–B*). As a control, we tested co-localization with

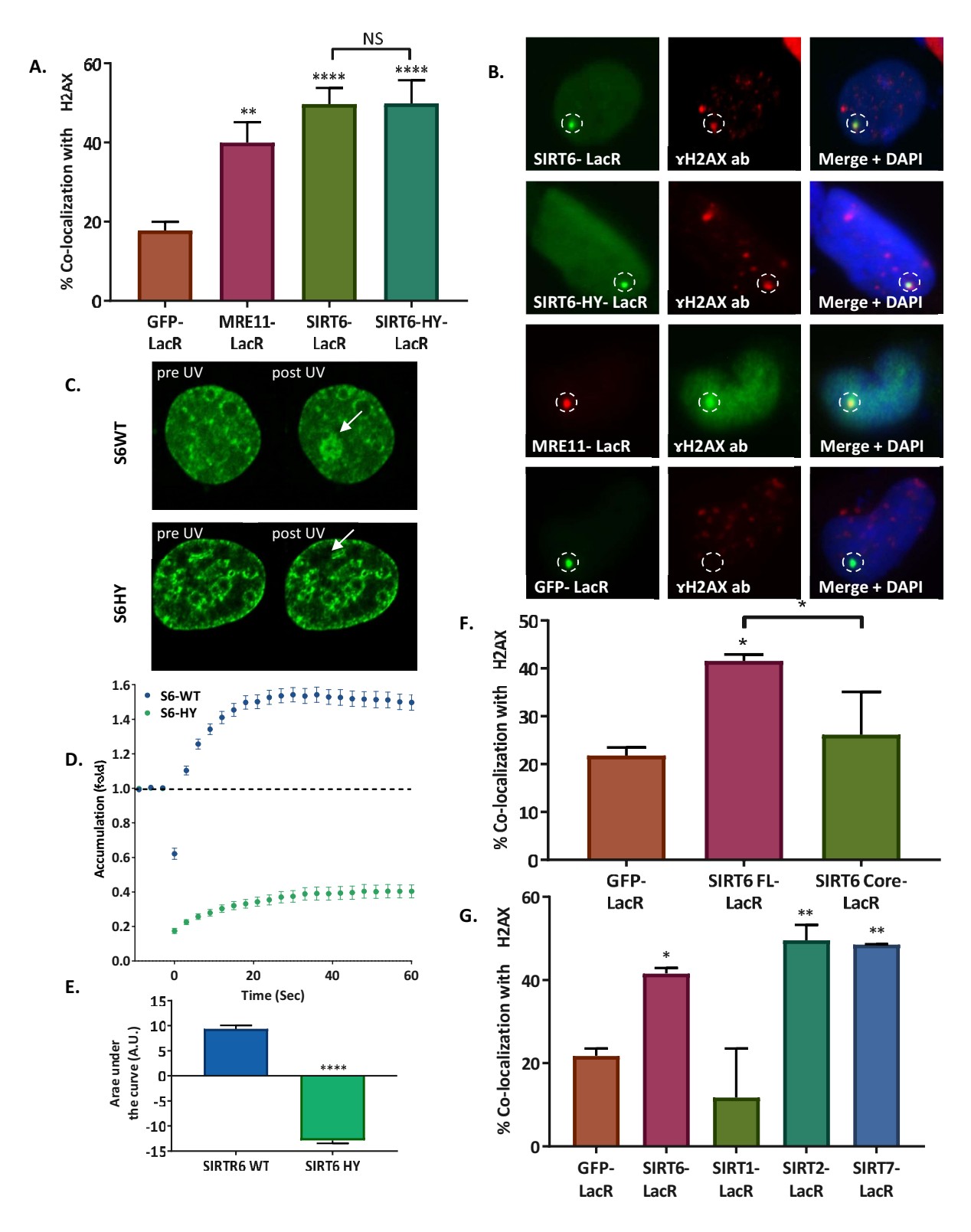

**Figure 5.** SIRT6 can initiate the DNA damage response. (A, B) Initiation of the DDR, measured by co-localization of MRE11-LacR-Cherry (n = 136, p<0.005), SIRT6-LacR-GFP (n = 243, p<0.0001) or SIRT6 HY-LacR-GFP (n = 71, p<0.0001), compared to GFP-LacR (n = 310). Data are means for 4–9 experiments (error bars are SEMs). (C, D) Live imaging upon laser-induced damage (LID) of SIRT6-WT-GFP (n = 20) or SIRT6-HY-GFP (n = 20) in SIRT6 KO U2OS cells (n = 20). (D) Accumulation over time in 3 s intervals. (E) Average area under the curve of for three LID experiments (error bars are SEMs)
*Figure 5 continued on next page*

*Figure 5 continued*

(n[S6-WT]=40, n[S6-HY]=39, p<0.0001). (**F**) Initiation of DDR by full-length SIRT6-LacR-GFP (n = 127, p<0.005), Core-LacR-GFP (n = 66, p>0.05) or GFP-LacR (n = 64). Data are averages from 3–4 experiments (error bars show SEMs). (**G**) Initiation of DDR by SIRT6-LacR-GFP (n = 127, p<0.05), SIRT1-LacR-GFP (n = 44, p>0.05), SIRT2-LacR-GFP (n = 67, p<0.005), SIRT7-LacR-GFP (n = 68, p<0.005) or GFP-LacR (n = 64). Data are averages from 3–10 experiments (error bars show SEMs).

The online version of this article includes the following figure supplement(s) for figure 5:

**Figure supplement 1.** SIRT6 can initiatetheDNA damage response.

CDT1, a nuclear protein that does not participate in the DDR. As expected, CDT1 was neither recruited by SIRT6 nor by GFP alone.

As SIRT6 DDR activation is independent of its catalytic activity, we further examined whether it is needed for DDR protein recruitment. Taking advantage of the tethering assay, we observed that both SIRT6-WT and SIRT6-HY recruited 53BP1 and BRCA1, meaning that the recruitment is independent of SIRT6 catalytic activity (*Figure 6—figure supplement 2A–B*).

53BP1 and BRCA1 can antagonize each other, and a change in their concentration within the cell may influence the recruitment capacity. Therefore, we used IF to test whether overexpression of these proteins results in a different outcome from that produced by the endogenous proteins. However, the results were very similar, suggesting that the recruitment is independent of the amount of protein in the cell, and that an additional layer of regulation would influence the recruitment (*Figure 6—figure supplement 2C–D*).

The tethering assay can detect both protein–protein interaction or recruitment through signaling. To differentiate these two possibilities, we inhibited DDR signaling by supplementing the media of the cells with Wortmannin, thus inhibiting ATM, ATR and DNA-PKc (scheme in *Figure 6—figure supplement 3A*). Our results indicate that when these kinases are inhibited (shown by a reduction in γH2AX levels), the recruitment of both 53BP1 and BRCA1 to the LacO site is reduced (*Figure 6—figure supplement 3B–D*). However, the recruitment of the DDR initiators Ku80 and MRE11 is not affected by Wortmannin, suggesting that their recruitment is based on protein–protein interactions and not on downstream signaling alone (*Figure 6—figure supplement 3E*). These results indicated that SIRT6 participates in DDR activation through the initiation of signaling and the recruitment of various proteins, which lead to the different DNA-repair pathways.

## Discussion

In this work, we discovered a novel function for the chromatin factor SIRT6 as a DSB sensor that is able to bind DSBs and initiate the cellular DDR.

We showed that SIRT6 can bind DNA with high affinity for ssDNA and open-ended dsDNA. We believe that the binding occurs through a tunnel-like structure in the protein core domain, close to its catalytic site. This structure could only fit ssDNA, and whereas other proteins require resection for ssDNA identification, 3–4 bases are enough for SIRT6.

By generating several point mutations in the hypothesized DNA-binding site, we managed to reduce the DNA-binding capacity of SIRT6, also reducing the catalytic activity. However, A13W and D63Y mutations raise the possibility that, despite the proximity of these sites, these abilities are distinct ones. The D63Y mutation had no effect on DNA binding, but it caused a significant reduction in SIRT6 catalytic activity. A13W mutation, on the other hand, resulted in an increase in catalytic activity along with a slight reduction in DNA binding.

In addition, we showed that SIRT6 can arrive at the sites of DSBs independently of the known sensors MRE11 and Ku80 and of PARP activity, and can activate the DDR on its own. We also observed that its catalytic activity is not necessary for DDR initiation when it is already bound to the DNA, as shown by the ability of SIRT6-HY-LacR to initiate the DDR. However, because the binding capacity in the HY mutant is reduced, we believe that SIRT6-HY is not able to bind and remain attached to the DNA (as shown by its impaired recruitment to laser-induced damage sites), and therefore that all DDR initiation would be impaired by this mutant. Interestingly, even though the initiation of the DDR occurs when SIRT6 is catalytically inactive, it cannot be initiated by the active core-domain alone. These results suggest a complex relationship between binding capacity and activation, in which binding per se cannot result in DDR signaling.

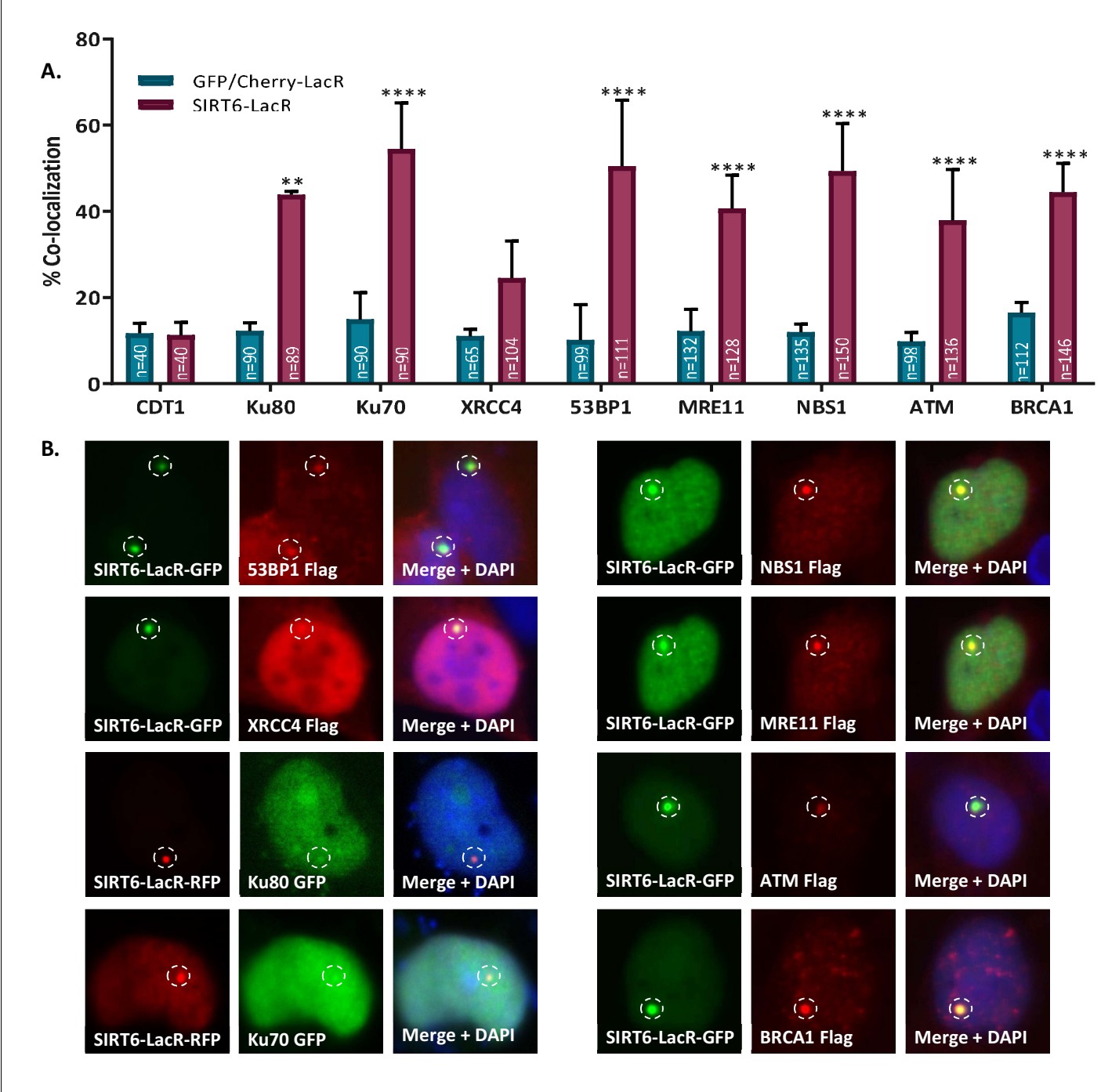

**Figure 6.** SIRT6 can recruit enzymes of both the NHEJ and HR repair pathways. (**A, B**) Percentage repair enzymes that are co-localized with SIRT6-LacR-GFP/Cherry at LacO sites. IF with Flag antibody. Data are averages for 3–6 experiments (with error bars representing SEMs) (*, p<0.05; **, p<0.005; ***, p<0.0005; ****, p<0.00005).

The online version of this article includes the following figure supplement(s) for figure 6:

**Figure supplement 1.** SIRT6 can recruit enzymes of boththeNHEJ and HR repair pathways.
**Figure supplement 2.** SIRT6 can recruit enzymes of boththeNHEJ and HR repair pathways.
**Figure supplement 3.** SIRT6 can recruit enzymes of boththeNHEJ and HR repair pathways.

**Table 1.** DNA sequences used in the EMSA assay.

**DNA sequences used in the EMSA assays**

| | | | |
|---|---|---|---|
| ssDNA | 5′ | GGGAAAGTTGACGGGAGGGTAT TGGAGGTTAGTGGAGGTGAGTGG | 3′ |
| ssDNA | 5′ | CCACTCACCTCCACTAACCTCC AATACCCTCCCGTCAACTTTCCC | 3′ |
| dsDNA-Blunt | 5′ | CCACTCACCTCCACTAACCTCC AATACCCTCCCGTCAACTTTCCC | 3′ |
| dsDNA-recessed | 5′ | CCACTCACCTCCACTAACCTCC AATACCCTCCCGTCAAC | 3′ |
| dsDNA-recessed | 5′ | ACCTCCACTAACCTCCAATACC CTCCCGTCAACTTTCCC | 3′ |
| dsDNA-Blunt | 5′ | AAGGTCGACACCACCTTTGAGAGCGC GCGGCCCACGCAGACCCACATGGCGCT GGTGCAGCTGGAGCGCGTGGGCCTCCT CCGCTTCCTGGTCAGCCAGAACGTCGACAAA | 3′ |
| dsDNA-recessed | 5′ | TCGACACCACCTTTGAGAGCGCGCGG CCCACGCAGACCCACATGGCGCTGGTGCAGC TGGAGCGCGTGGGCCTCCTCCGC TTCCTGGTCAGCCAGAACG | 3′ |
| RNA | 5′ | GCGAAGUCUUCGU | 3′ |

Given that the core domain, which contains both the catalytic domain and the DNA-binding domain of SIRT6, is conserved among Sirtuins, we also showed that other Sirtuins share the ssDNA-binding capacity (but with different affinities). This is especially interesting as Sirtuins are present in the cell at different cellular locations (cytoplasm, nucleus and mitochondria) and have different catalytic activities (deacetylases, deacylases, and ADP ribosylases) (*Liszt et al., 2005*). This suggests that the DSB-binding capacity could be relevant in other cellular compartments, for example, in mitochondrial DNA repair. When nuclear SIRT2 and SIRT7 were forced to localize to the DNA by the LacO-LacR tethering assay, they were also able to initiate the DDR. However, SIRT7 lacks the broken-DNA binding specificity and SIRT2 has a poor binding capacity, which would impair their roles as DNA damage sensors.

These findings open new possibilities for the cellular functions of the Sirtuin family; nevertheless, we believe in the uniqueness of SIRT6 as it possesses all of these abilities at once.

The placing of SIRT6 as a sensor of DSBs might explain why the lack of SIRT6 gives rise to one of the most striking phenotypes in humans, monkeys and mice, including phenotypes that are typically associated with genomic instability such as premature ageing, accelerated neurodegeneration, tissue atrophy and cancer (*Kugel and Mostoslavsky, 2014*; *Tasselli et al., 2017*). In particular, SIRT6 is involved in several repair pathways. As a sensor and DDR initiator, its absence would have deleterious effects on the whole downstream DDR signaling. Our results point out that its role begins as a DSB sensor (although it may recognize other DNA lesions), recognizing and initiating the DDR independently of other factors. SIRT6 has multiple functions in the context of chromatin (*Kugel and Mostoslavsky, 2014*), including transcriptional regulation. Thus, it might seem somewhat paradoxical that it can initiate the DDR response by merely binding to damage sites.

It is not particularly clear how SIRT6 can selectively activate the DDR when bound to DNA damage sites but not when bound to sites of transcription regulation. A possible explanation could rely on the fact that transcription factors are very dynamic, and they usually bind chromatin transiently (*Hager et al., 2009*). Therefore, we speculate that SIRT6, similarly to MRE11, probes the DNA transiently, and that even though it is constantly present in chromatin, its binding to unbroken DNA is not as tight as when it encounters broken DNA (as seen in the binding assays) (*Myler et al., 2017*). Tighter binding of SIRT6 might allow stabilization through protein interactions and modifications, analogous to the processes that occur with MRE11, NBS1, ATM and other DDR proteins. It is also possible that SIRT6 undergoes a conformational change when bound to broken DNA. However, our tethering system suggests that its continuous presence in chromatin (in the absence of broken DNA to bind) is sufficient to initiate the DDR cascade.

**Table 2.** Theoretical Rg (Å) derived from the SAXS data.

| Theoretical Rg (Å) | |
| --- | --- |
| SIRT6 dimer | 27.14 |
| SIRT6 tetramer | 36 |
| SIRT6 hexamer | 40 |

Interestingly, unlike other factors, SIRT6 recruitment and kinetics are not affected by PARP activity, making it independent of PARylation and giving it an advantage over other factors that require PARylation for fast recruitment (*Mao et al., 2011*). This feature could be relevant as an adjuvant therapy in cancer treatment (*Beck et al., 2014*; *Haince et al., 2008*).

It is also important to note that although SIRT6 can recruit proteins of both HR and NHEJ and its deficiency affects both pathways, SIRT6 KO impaired the recruitment of MRE11, but not Ku80, to sites of laser-induced damage. It is possible that the Ku complex does not require SIRT6 for recognition, yet it may require SIRT6 chromatin remodeling activity in later repair steps as NHEJ repair is affected by the lack of SIRT6. Alternatively, as in the case of PARP1 and the MRN complex, SIRT6 might compete with the Ku complex for DSB binding and DDR initiation (*Myler et al., 2017*; *Yang et al., 2018*) .

Our findings place SIRT6 at the beginning of the DDR response as a novel DSB sensor, but how it affects the DSB repair pathway choice still needs to be investigated. Nevertheless, as there is significant cross-talk between the pathways (seen, for example, by the involvement of the HR initiator MRE11 in NHEJ [*Xie et al., 2009*]), it is possible that it has roles in both.

In conclusion, we have demonstrated that SIRT6 has a role as an independent DNA damage sensor. This is critical for the initiation of the DSB-DNA damage response and hence for the support of genomic stability and health.

# Materials and methods

## Key resources table

| Reagent type (species) or resource | Designation | Source or reference | Identifiers | Additional information |
| --- | --- | --- | --- | --- |
| Recombinant DNA reagent | ATM-LacR-Cherry | *Soutoglou and Misteli, 2008* | | |
| Recombinant DNA reagent | CDT1-TagRFP | ThermoFisher | P36237 | |
| Recombinant DNA reagent | CMV-Flag | *Toiber et al., 2013* | | |
| Recombinant DNA reagent | MRE11-Flag | *Wu et al., 2008* | | |
| Recombinant DNA reagent | MRE11-LacR-Cherry | *Soutoglou and Misteli, 2008* | | |
| Recombinant DNA reagent | mRFP-SIRT6 | *Kaidi et al., 2010* - retracted | | |
| Recombinant DNA reagent | NBS1-Flag | *Wu et al., 2008* | | |
| Recombinant DNA reagent | | NBS1-LacR-Cherry | | |
| Recombinant DNA reagent | pcDNA3.1(+)Flag -His-ATM-WT | Addgene | 31985 | |
| Recombinant DNA reagent | pcDNA5-FRT/T0-Flag-53BP1 | Addgene | 52507 | |
| Recombinant DNA reagent | pDEST 3x Flag-pcDNA5-FRT/T0-BRCA1 | Addgene | 52504 | |

*Continued on next page*

*Continued*

| Reagent type (species) or resource | Designation | Source or reference | Identifiers | Additional information |
|---|---|---|---|---|
| Recombinant DNA reagent | pEGFP-C1-FLAG-Ku70 | Addgene | 46957 | |
| Recombinant DNA reagent | pEGFP-C1-FLAG-Ku80 | Addgene | 46958 | |
| Recombinant DNA reagent | pEGFP-C1-FLAG-XRCC4 | Addgene | 46959 | |
| Recombinant DNA reagent | pEGFP- SIRT6 | *Kaidi et al., 2010* - retracted | | |
| Recombinant DNA reagent | pET28 hSIRT6-His | *Gertman et al., 2018* | | |
| Recombinant DNA reagent | pHPRT-DRGFP | Addgene | 26476 | |
| Recombinant DNA reagent | pMal-C2-hSIRT6-WT | *Gertman et al., 2018* | | |
| Recombinant DNA reagent | pMal-C2-hSIRT6-A13W | This paper | | |
| Recombinant DNA reagent | pMal-C2-hSIRT6-D63H | This paper | | |
| Recombinant DNA reagent | pMal-C2-hSIRT6-D63Y | This paper | | |
| Recombinant DNA reagent | pMal-C2-hSIRT6-H133Y | *Gertman et al., 2018* | | |
| Recombinant DNA reagent | pMal-C2-hSIRT6-D188A | This paper | | |
| Recombinant DNA reagent | pMal-C2-hSIRT6-D190W | This paper | | |
| Recombinant DNA reagent | pMal-C2-hSIRT6-I217A | This paper | | |
| Recombinant DNA reagent | pQCXIP-Cherry-LacR | This paper | | |
| Recombinant DNA reagent | pQCXIP-SIRT6 Core-GFP-LacR | This paper | | |
| Recombinant DNA reagent | pQCXIP-GFP-LacR | Addgene | 59418 | |
| Recombinant DNA reagent | pQCXIP-KU80-GFP-LacR | This paper | | |
| Recombinant DNA reagent | pQCXIP-mSIRT6-Cherry-LacR | This paper | | |
| Recombinant DNA reagent | pQCXIP-mSIRT6-GFP-LacR | This paper | | |
| Recombinant DNA reagent | pQCXIP-mSIRT6-H133Y-GFP-LacR | This paper | | |
| Recombinant DNA reagent | pQCXIP-SIRT1-GFP-LacR | This paper | | |
| Recombinant DNA reagent | pQCXIP-SIRT2- GFP-LacR | This paper | | |
| Recombinant DNA reagent | pQCXIP-SIRT7- GFP-LacR | This paper | | |
| Recombinant DNA reagent | SIRT1-Flag | *Zhong et al., 2010* | | |
| Recombinant DNA reagent | SIRT2-Flag | Addgene | 13813 | |

*Continued on next page*

*Continued*

| Reagent type (species) or resource | Designation | Source or reference | Identifiers | Additional information |
|---|---|---|---|---|
| Recombinant DNA reagent | SIRT3-Flag | Addgene | 13814 | |
| Recombinant DNA reagent | SIRT4-Flag | Addgene | 13815 | |
| Recombinant DNA reagent | SIRT5-Flag | Addgene | 13816 | |
| Recombinant DNA reagent | SIRT6 Core | *Tennen et al., 2010* | | |
| Recombinant DNA reagent | mSIRT6-WT-Flag | *Zhong et al., 2010* | | |
| Recombinant DNA reagent | SIRT7-Flag | Addgene | 13818 | |
| Recombinant DNA reagent | SNF2H-WT-GFP-LacR | *Klement et al., 2014* | | |
| Antibody | Alexa Fluor 488 AffiniPure Donkey Anti-Rabbit IgG (H+L) | Jackson Immunoresearch | 711-545-152 | IF (1:200) |
| Antibody | Alexa Fluor 594 AffiniPure Donkey Anti-Rabbit IgG (H+L) | Jackson Immunoresearch | 711-585-152 | IF (1:200) |
| Antibody | Alexa Fluor 488 AffiniPure Donkey Anti-Mouse IgG (H+L) | Jackson Immunoresearch | 715-545-150 | IF (1:200) |
| Antibody | Alexa Fluor 594 AffiniPure Goat Anti-Mouse IgG (H+L) | Jackson Immunoresearch | 115-585-062 | IF (1:200) |
| Antibody | 53 BP1 | Snata-Cruz Biotechnology | sc-22760 | IF (1:300) |
| Antibody | BRCA1 | Snata-Cruz Biotechnology | sc-7298 | WB (1:1000) |
| Antibody | Flag | Sigma-Aldrich | F1804 | IF (1:900), WB (1:1000) |
| Antibody | Flag Beads | Sigma-Aldrich | A2220 | IP |
| Antibody | gamma H2A.X (phospho s139) | abcam | ab2893 | IF(1:3000), WB (1:1000) |
| Antibody | Goat Anti-Rabbit IgG H and L (HRP) | abcam | ab6721 | WB (1:10000) |
| Antibody | Histone H3 | abcam | ab1791 | WB (1:5000) |
| Antibody | Histone H3 (acetyl K56) | abcam | ab76307 | WB (1:1000) |
| Antibody | HSC 70 | Snata-Cruz Biotechnology | sc-7298 | WB (1:1000) |
| Antibody | Ku80 | Cell Signaling | #2180 | WB (1:1000) |
| Antibody | MRE11 | abcam | ab214 | WB (1:1000) |
| Antibody | phospho-ATM (Ser1981) | Cell Signaling | #5883 | WB (1:1000) |
| Antibody | phospho-Histone H2A.X (Ser139) | Millipore | 05–636 | IF (1:1500) |
| Antibody | Rabbit Anti-Mouse IgG H and L (HRP) | abcam | ab97046 | WB (1:10000) |

*Continued on next page*

*Continued*

| Reagent type (species) or resource | Designation | Source or reference | Identifiers | Additional information |
|---|---|---|---|---|
| Antibody | SIRT6 | abcam | ab62739 | WB (1:1000) |
| Antibody | Tubulin | Merck | MAB1637 | WB (1:1000) |
| Peptide, recombinant protein | SIRT1 Human | PROSPEC | PRO-1909 | DNA binding assay |
| Peptide, recombinant protein | SIRT3 Human | PROSPEC | PRO-462 | DNA binding assay |
| Peptide, recombinant protein | SIRT5 Human | PROSPEC | PRO-1774 | DNA binding assay |
| Peptide, recombinant protein | SIRT6 Human | PROSPEC | PRO-282 | DNA binding assay |

## Cell cultures

All cells were cultured in DMEM and 4.5 g/l glucose, supplemented with 10% fetal bovine serum, 1% penicillin and streptomycin cocktail and 1% L-glutamine. Cells were cultured with 5% $CO_2$ at 37° C.

All lines were confirmed to be *Mycoplasma*-free using a hylabs Hy-mycoplasma Detection PCR Kit with internal control (Cat No. KI 5034I).

Cells were authenticated by the Biochemical Core Facility of the Genomics Center at the Technion-Israel Institute of Technology.

## Plasmids and transfections

To prepare pQCXIP-msirt6-GFP-LacR, mouse *sirt6* without a stop codon was amplified by PCR and introduced in frame with GFP-LacR into the AgeI site of plasmid pQCXIP-GFP-LacR (Addgene, 59418).

pQCXIP-mSIRT6-H133Y-GFP-LacR was prepared by Quick Change Site-directed mutagenesis of mSIRT6 flanked by AgeI sites in pGEM, and after sequencing, introduced to the AgeI site in frame with the fused GFP-LacR of pQCXIP-GFP-LacR (Addgene, 59418).

pQCXIP-Cherry-LacR was prepared by excision of the AgeI/XhoI GFP fragment of pQCXIP-GFP-LacR and exchanged with AgeI/XhoI mCherry amplified from pDEST-mCherry-LacR-BRCA1 (Addgene, 71115).

pQCXIP-mSIRT6-Cherry-LacR was prepared by introducing the AgeI mSIRT6 from pQCXIP-KU80-GFP-LacR and by introducing KU80, amplified from pEGFP-C1-FLAG-Ku80 (Addgene, 46958), into the AgeI site of pQCXIP-GFP-LacR in frame with GFP.

pQCXIP-hSIRT1-GFP-LacR was prepared by inserting the amplified SIRT1 from SIRT1-Flag (Mostoslavsky Lab) with the AgeI site in frame with the GFP-LacR of plasmid pQCXIP-GFP-LacR (Addgene, 59418).

pQCXIP-hSIRT2-GFP-LacR was prepared by inserting the amplified SIRT2 from SIRT2-Flag (Addgen #13813) with the AgeI site in frame into the GFP-LacR of plasmid pQCXIP-GFP-LacR (Addgene, 59418).

pQCXIP-hSIRT7-GFP-LacR was prepared by inserting the amplified SIRT7 from SIRT7-Flag (Addgen #13818) with the AgeI site in frame into the GFP-LacR of plasmid pQCXIP-GFP-LacR (Addgene, 59418).

pQCXIP-Core hSIRT6-GFP-LacR was prepared by inserting the amplified 233 amino acid (aa) core region from aa 43 to aa 276 of human SIRT6 and introducing it into the AgeI site of pQCXIP-GFP-LacR (Addgene, 59418) with an additional methionine before aa 43 and in frame with the GFP-LacR of the plasmid.

pMal-C2-hSIRT6 A13W, D63H, D63Y, W188A, D190Wand I217A were prepared by Quick Change Site-directed Mutagenesis on pMal-C2-hSIRT6. The mutation was affirmed by sequencing.

All PCRs were performed with Hot start, KAPA HiFi #KM 2605 or abm Kodaq #G497-Dye proof-reading polymerases. All clones were sequenced for validation, and expression of the fluorescent fusion proteins were checked by transfection into cells. All transfections were performed using PolyJet In Vitro Transfection (SignaGen, SL100688), according to the manufacturer's instructions.

## Immunofluorescence

U2OS cells were washed with PBS and fixed with 2% paraformaldehyde for 15 min at room temperature, followed by an additional wash. Quenching was then performed with 100 mM glycine for 5 min at room temperature (RT). Cells were permeabilized (0.1% sodium citrate, 0.1% Trition X-100 [pH 6], in deionized distilled water [DDW]) for 5 min and washed again. After 1 hr blocking (0.5% BSA, 0.1% Tween-20 in PBS), cells were incubated with primary antibody diluted in blocking buffer over night at 4°C. The next day, cells were washed three times with wash buffer (0.25% BSA, 0.1% Tween-20 in PBS), incubated for 1 hr with secondary antibody (diluted in blocking buffer 1:200) at RT and washed three more times. Cells were then DAPI stained for three minutes at RT and washed with PBS twice before imaging.

## Tethering assay

U2OS cells containing 256X LacO sequence repeats in their genome were transfected with plasmids of chimeric LacR-DDR enzyme-GFP/Cherry proteins. Cells were either co-transfected with a second plasmid of a fluorescent/Flag-tagged protein or immuno-stained (see 'Immunofluorescence') for an endogenic protein.

Cells expressing both proteins of interest and exhibiting visible foci of LacR-DDR-GFP/Cherry at LacO sites were located using an Olympus IX73 fluorescent microscope, whereas co-localization between both proteins was assessed visually using Olympus CellSens Software. Co-localization is defined as the common localization of large foci of the two proteins of interest at the LacO site. Co-localization was assessed as either positive (1) or negative (0). From this analysis, the percentage of cells that exhibit co-localization (positive cells) was calculated, and defined as 'percentage of co-localization between two proteins'. The co-localization percentage for each protein of interest was compared to the co-localization percentage with LacR-GFP/Cherry as a control.

Notes: the pQCXIP-Ku80-GFP-LacR plasmid used in this assay contains Ku80 that was acquired from Addgene (cat. #46958) and contains the D158G mutation.

The pQCXIP-SIRT1-GFP-LacR plasmid used in this assay contains SIRT1 that was obtained from the Mostoslavsky lab (*Zhong et al., 2010*). This protein variant is lacking 79 amino acids in the N-terminus.

## Immunoprecipitation (IP)

Flag-tagged proteins were purified from transfected HEK293T cells. Cells were collected and washed with PBS. Cell disruption was performed in lysis buffer (0.5M KCl, 50 mM Tris-HCl [pH 7.5], 1% NP40, 0.5M DTT, 200 mM TSA and protease and phosphatase inhibitors in DDW) by 10 min rotation at 4°C. Cell debris were sedimented by 15 min centrifugation at 21,000 g. Lysate was collected and added to ANTI-FLAG M2 Affinity Gel (Sigma-Aldrich, A2220) beads for 2 hr rotation at 4°C. Beads were then washed three times with lysis buffer and once with SDAC buffer (50 mM Tris-HCl [pH 9], 4 mM MgCl, 50 mM NaCl, 0.5 mM DTT, 200 mM TSA and protease and phosphatase inhibitors in DDW). Proteins were released by flag-peptide.

## Expression and purification of recombinant SIRT6 in *Escherichia coli*

Expression and purification of His-tagged and MBP-tagged proteins in *E. coli* were performed as previously described by *Gertman et al. (2018)*.

## Fluorescence recovery after photobleaching (FRAP)

FRAP experiments (laser-induced damage) were performed as previously described by *Toiber et al. (2013)*. In brief, cells were plated in Ibidi μ-Slide eight-well glass bottom plates (Cat. No.: 80827) and transfected with the desired fluorescent plasmid. Pre-sensitization with Hoechst (1 mM) was done for 10 min before the experiment. FRAP experiments were carried out using a Leica SP5 microscope (German Cancer Research Center (DKFZ) and BioQuant, Heidelberg, Germany) or using a

LSM880 microscope (Ben Gurion University, Be'er Sheva, Israel) with a 63X oil immersion objective. Images were acquired in a 512 × 512 format with a scan speed of 1,400 Hz. Circular bleach spots of 2 µm diameter were positioned either at a damage site or at a distant reference site. Spots were bleached with an argon laser of 488 nm with a power of 1 mW in the back aperture of the objective. Images were taken at 3 s intervals, with three baseline images taken before bleaching. Acquisition before bleaching was used for normalization of each cell intensity (average of the baseline intensity of the whole cell nucleus prior to DNA damage). Images analysis and fluorescence assessment were performed using ImageJ 1.52i software.

To assess protein amounts and to compare between the different conditions, area under the curve was calculated using a MATLAB pipeline.

## DNA-binding assay

Open-ended plasmids were prepared in advance by incubating DR-GFP plasmids with EcoRV for blunt ends, KpnI for 3' over hang or SalI for 5' over hang according to manufacture instructions. Circular plasmids were subjected to the same conditions with no restriction enzyme.

To achieve protein–DNA binding, flag-tagged proteins that were previously immunoprecipitated were incubated at 37°C for 1 hr with same amount of circular or open-ended DNA, 1:5 of 5X deacetylation buffer (50 mM Tris HCl [pH 8], 50 mM NaCl, 4 mM MgCl$_2$ and 0.5 mM DTT in DDW), and 1:50 50X protease inhibitors in DDW.

ANTI-FLAG M2 Affinity Gel (Sigma-Aldrich, A2220) beads were blocked with 5% BSA supplemented with 1X deacetylation buffer (with 1% phosphate inhibitors) by rotation for 1 hr in 4°C. Beads were then centrifuged (1000 g, 3 min, 4°C) and buffer was changed to clean deacetylation buffer 1X. Beads were then distributed equally between all samples.

To achieve beads–protein binding, samples were rotated for 2 hr in 4°C. After rotation, samples were centrifuged (1000 g, 3 min, 4°C) and washed 3 times with 1 ml of wash buffer (0.1% SDS, 0.5% Triton x-100, 2 mM EDTA, 20 mM Tris-HCl [pH8] and 150 mM NaCl in DDW).

Protein–DNA complexes were then released by two rounds of 20 min vortexing at room temperature with 100 µl elution buffer (0.1M NaHCO$_3$ and 1% SDS in DDW).

For His-tagged proteins (acquired from PROSPEC), the assay was performed using HisPur Ni-NTA Resin (ThermoFisher, 88221) under the same conditions with the appropriate buffers (binding buffer — 20 mM Tris HCl [pH 8], 150 mM NaCl, 10% PMSF, 1% phosphatase inhibitors; wash buffer — 20 mM Tris HCl [pH 8], 150 mM NaCl, 20 mM imidazole; elution buffer — 20 mM Tris HCl [pH 8], 150 mM NaCl, 500 mM imidazole).

Notes: the SIRT1-Flag used in this assay was obtained from the Mostoslavsky lab (PDMI: 20141841). This protein variant is lacking 79 amino acids in the N-terminus.

The SIRT1-His used in this assay was acquired from PROSPEC (https://www.prospecbio.com/sirt1_human). This SIRT1 is a 280 aa poly-peptide (aa 254–495).

## DNA isolation

1:1 vol of phenol:chloroform:isoamyl alcohol (25:24:1) was added to the eluted DNA from the DNA binding assay, vortexed and centrifuged at RT for 5 min at 17,000 g. The top aqueous layer was then isolated and washed with 1 vol of chloroform: isoamyl alcohol (24:1). Samples were then centrifuged under the same conditions, and the top aqueous layer was isolated. 1/10 vol 3M NaOAc, 30 µg glycogen and 2.5 volumes ice cold 100% EtOH were added to each sample, followed by incubation for at least 30 min at −80°C. After incubation, DNA was precipitated by centrifugation at max. speed for 30 min at 4°C, supernatant was discarded and the pellet was washed with 500 µl 70% ice-cold EtOH. Samples were then centrifuged at max. speed for 30 min at 4°C, before the supernatant was discarded and the DNA pellet was air dried before re-suspension with ultra-pure water.

## Quantitative PCR

For relative quantification of the DNA isolated from all of the DNA-binding assays performed, qPCR was performed using SsoAdvanced Universal SYBER Green Supermix (BIO-RAD, 172–5274) according to the manufacturer's instructions.

Primers used for DR-GFP plasmid amplification:

Forward: 5'-TCTTCTTCAAGGACGACGACGGCAACT-3'

eLIFE Research article

Reverse: 5'-TTGTAGTTGTACTCCAGCTTGTGC-3'

## Exonuclease assay

DR-GFP plasmid was cut with restriction enzymes generating linear DNA with blunt (EcoRV) or over-hanging ssDNA (SalI or KpnI). DNA cleavage was confirmed by agarose gel electrophoresis. 10 μg of the restricted DNA was incubated with BSA, NBS1, MRE11 or SIRT6 purified proteins in NEB exonuclease buffer for 0 to 20 min. ExoI was then added to the samples. Samples of each reaction were taken at 0, 10 and 20 min. DNA was purified using a Qiagen PCR purification kit. The purified DNA was run on 0.8% agarose gel, and the amount of DNA was assessed by image analysis using ImageJ 1.52i software and normalized to the amount of the DNA at the 0' time point.

## Fluor de lys (FDL) activity assay

Fluor de lys assay with SIRT6-point mutant-MBP proteins was performed as previously described by *Gertman et al. (2018)*.

## NAD⁺ consumption assay

Purified SIRT6-Flag was incubated at 37 °C for 3 hr with either PstI digested pDR-GFP (DSB), ssDNA or a H3K56 acetylated peptide with 2.5 mM NAD⁺ and HEPES buffer (50 mM HEPES [pH 7.5], 100 mM KCl, 20 mM $MgCl_2$, and 10% glycerol). After incubation, samples were supplemented with 1 μM 1,3-propanediol dehydrogenase (1,3-PD) and 170 mM 1,3-propanediol for an additional 3 hr incubation. NAD⁺ consumption by SIRT6 was assessed by NADH levels produced by 1,3-PDase activity, by measuring its absorption at 340 nm. To monitor spontaneous NAD⁺ consumption in the presence of PstI digested pDR-GFP, ssDNA or H3K56 acetylated peptide, the assay was conducted without SIRT6, and each treatment was normalized to its control.

## EMSA

SIRT6 in storage buffer (20 mM Tris-HCl [pH 7.4], 150 mM NaCl and 50% glycerol) was equilibrated with DNA (or RNA) for 20 min on ice. The buffer composition of EMSA was optimized to obtain the maximum resolution for resolving DNA/RNA. Reactions (final volume 10 μL) were resolved by electrophoresis at 4°C through native gel containing 5% (for blunt-end and sticky-end DNA), 8% (for ssDNA) and 10% (for RNA) polyacrylamide (29:1 acrylamide: bisacrylamide) in 1X TBE buffer. Autoradiographs of the dried gels were analyzed by densitometry using Fujifilm PhosphorImager. The signal was quantified by ImageQuant TL.

GraphPad Prism 7 was used to estimate apparent Kd value for ssDNA (one site, specific binding fit, $y = B_{max}[SIRT6]/(K_d + [SIRT6])$ and for blunt-end and sticky-end DNA (specific binding with Hill slope, $y = B_{max}[SIRT6]^h/(K_d{}^h + [SIRT6]^h)$.

## SAXS

SAXS data were collected at BioSAXS beamline BM29 (ESRF, Grenoble, France), possessing a Pilatus 1M detector. The scattering intensity was recorded in the interval $0.0035 < q < 0.49$ Å$^{-1}$. The measurements were performed at 20°C. SIRT6 (alone or in the presence of dsDNA) was measured at a concentration of 0.5 mg/ml, as it tends to aggregate at higher concentrations. The scattering of the buffer was also measured and subtracted from the scattering of the samples by using Primus (*Konarev and Svergun, 2018*).

*Konarev and Svergun (2018)* PyMOL (https://pymol.org/) was used to extract the structures of the SIRT6 dimer and tetramer from the available crystal structure (PDB ID code: 3pki). CRYSOL (*Svergun et al., 1995*) was then used to compute the artificial SAXS spectra of each protein species. These spectra served as a reference for the reconstitution of experimental SAXS data.

Values for the radius of gyration ($R_g$) and the maximum particle dimension ($D_{max}$) were derived from distance distribution function *P(r)*, using in-house script (*Akabayov et al., 2010*). This script was designed to perform an automatic search for the best fitting parameters in GNOM (*Svergun, 1992*). In the end, DAMMIN (*Svergun, 1999*) was used to reconstruct the molecular envelope on the basis of the best GNOM fit (obtained from the script analysis and refined manually). E models were calculated and averaged using DAMMAVER (*Volkov and Svergun, 2003*).

## SEC-MALS

A miniDAWN TREOS multi-angle light scattering detector, with three detector angles (43.6°, 90° and 136.4°) and a 658.9 nm laser beam (Wyatt Technology, Santa Barbara, CA), with a Wyatt QELS dynamic light scattering module for determination of hydrodynamic radius and an Optilab T-rEX refractometer (Wyatt Technology), were used in-line with a size exclusion chromatography analytical column, Superdex 200 Increase 10/300 GL (GE, Life Science, Marlborough, MA) equilibrated in buffer (50 mM tris, 150 mM NaCl and 4 mM $MgCl_2$ [pH 8.0]).

Experiments were performed using an AKTA explorer system with a UV-900 detector (GE), at 0.8 ml/min. All experiments were performed at RT (25°C).

Data collection and mass calculation by SEC-MALS analysis were performed with ASTRA 6.1 software (Wyatt Technology). The refractive index of the solvent was defined as 1.331 and the viscosity was defined as 0.8945 cP (common parameters for PBS buffer at 658.9 nm). dn/dc (refractive index increment) value for all samples was defined as 0.185 mL/g (a standard value for proteins). For the SIRT6 experiment, 150 ul 4.5 mg/ml human-SIRT6-His was injected. For SIRT6+DNA, 200 μl human-SIRT6-His + 50 ul DNA was injected after 1 hr incubation at 37°C.

## Statistical analysis

Statistical analysis was done using GraphPad Prism 7. Analysis included either one-way or two-way ANOVA followed by a post-hoc Dunnet test or a Tukey test, respectively. Significance was set at $p < 0.05$.

For all DNA binding assay results, statistical analysis was preceded by logarithmic transformation to overcome large variance between the different experiments. Statistical analysis was performed on the transformed data as described.

## Acknowledgements

This work was supported by ISF 188/17 and by the High-tech, Bio-tech and Chemo-tech scholarship of Kreitman School of Advanced Research of Ben Gurion University. We appreciate the plasmids kindly donated by Prof. Misteli and the U20S-LacO cells from Prof. Greenberg. We thank the staff scientist of beamline BM29 of ESRF (Grenoble, France) for providing support and the Israeli Block Allocation Group (BAG) for providing access. We thank Dr Mario Lebendiker and Dr Hadar Amartely from the Protein Purification Facility Wolfson Centre for Applied Structural Biology - The Hebrew University of Jerusalem, for their help with the SEC MALS experiments. We thank Prof. Eyal Gur and Dr Maayan Korman from Ben-Gurion University for their contribution to the development of the $NAD^+$ consumption assay. We thank Prof. Amir Aharoni and Dr Adi Hendler from Ben-Gurion University for their help and advice.

## Additional information

### Funding

| Funder | Grant reference number | Author |
| --- | --- | --- |
| Israel Science Foundation | 188/17 | Debra Toiber |
| Ben Gurion University | High-tech, Bio-tech and Chemo-tech scholarship | Debra Toiber |

The funders had no role in study design, data collection and interpretation, or the decision to submit the work for publication.

### Author contributions

Lior Onn, Conceptualization, Formal analysis, Investigation; Miguel Portillo, Fabian Erdel, Formal analysis, Investigation; Stefan Ilic, Investigation, Methodology; Gal Cleitman, Daniel Stein, Shai Kaluski, Ido Shirat, Zeev Slobodnik, Investigation; Monica Einav, Methodology, Project administration; Barak Akabayov, Formal analysis, Investigation, Methodology; Debra Toiber, Conceptualization, Formal analysis, Funding acquisition, Investigation

## Author ORCIDs

Fabian Erdel (iD) https://orcid.org/0000-0003-2888-7777
Barak Akabayov (iD) http://orcid.org/0000-0002-3882-2742
Debra Toiber (iD) https://orcid.org/0000-0002-1465-0130

## Decision letter and Author response

Decision letter https://doi.org/10.7554/eLife.51636.sa1
Author response https://doi.org/10.7554/eLife.51636.sa2

## Additional files

### Supplementary files

• Transparent reporting form

### Data availability

All the data generated or analyzed during this study are included in the manuscript and supporting files.

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
