## [Decision Letter]

**Acceptance summary:**

This study by Onn and colleagues uncovers a new role of the sirtuin enzyme SIRT6 as a DNA damage sensor, which can directly bind to DNA double-strand breaks (DSBs) and is critical for initiating cellular DNA damage responses. The findings support the model that SIRT6 functions very early after DNA damage and precedes commitment to particular DNA repair pathways. The study also identifies a tunnel-like structure within the SIRT6 protein, which may be responsible for recognizing broken DNA ends. SIRT6 is of great interest in aging biology; overexpression of SIRT6 in mice can extend lifespan, and SIRT6-mediated modifications of histones and other proteins regulate key signaling pathways relevant for cellular and organismic homeostasis. SIRT6 has previously been implicated in promoting DNA repair via multiple mechanisms, including recruiting DNA repair factors to sites of DNA damage and interacting with specific chromatin factors. The new conclusions in this study add to this area and improve the granularity of how we understand SIRT6 function in DNA repair processes. The work provides important novel contributions to the molecular understanding of SIRT6 function and the study of DNA damage repair.

**Decision letter after peer review:**

[Editors’ note: the authors submitted for reconsideration following the decision after peer review. What follows is the decision letter after the first round of review.]

Thank you for submitting your work entitled "SIRT6 is a DNA Double-Strand Break Sensor" for consideration by *eLife*. Your article has been reviewed by three peer reviewers, one of whom is a member of our Board of Reviewing Editors, and the evaluation has been overseen by a Senior Editor. The reviewers have opted to remain anonymous.

Our decision has been reached after consultation between the reviewers. Based on these discussions and the individual reviews below, we regret to inform you that your work will not be considered further for publication in *eLife* at this time. However, if you are able to address in full the reviewers' concerns in a substantially revised manuscript, we do encourage you to consider resubmitting to *eLife*.

The consensus of the reviewers is that the study proposes interesting points that would in principle be of sufficient novelty to merit publication in *eLife*, including the very early role of SIRT6 in the DDR and the direct binding of SIRT6 to broken DNA. However, there were major technical concerns, detailed in the accompanying reviews, which were assessed as not possible to address within the 2 month revision window of *eLife*. These include, but are not limited to: (1) the lack of essential controls demonstrating full inactivation of ATM or RNF8; (2) apparently conflicting data and insufficient methodological detail in SIRT6 localization analyses; (3) unconvincing data failing to fully support conclusions from tunnel mutations studies and inhibition of SIRT6 catalytic activity. In addition, the manuscript would benefit from considerable editing, as there are numerous areas with grammatical or other linguistic errors. Thus, the reviewers' concerns are substantial. However, if the authors can fully address all of these concerns with new experimental work, we would be open to consider the work in a new submission.

*Reviewer #1:*

This manuscript by Onn and colleagues presents evidence that the sirtuin SIRT6 is able to directly bind to DNA to initiate double-strand break (DSB) responses. SIRT6 is of great interest in aging biology; systemic overexpression of SIRT6 extends mouse lifespan, and it has been reported to regulate several key signaling pathways relevant for cellular and organismic homeostasis. SIRT6 has been implicated in promoting DNA repair via multiple mechanisms: e.g. activating PARP, promoting DNA-PK localization to sites of DNA damage, interaction with specific chromatin factors, etc. The conclusions of the authors, if fully substantiated, would add significantly to this area, and represent important novel contributions to sirtuin biology and the study of DNA damage repair. However, there are important areas where additional experimentation is needed to fully substantiate the conclusions. Specifically:

1) For Olaparib studies, the authors need to show that PAR levels are diminished under their treatment conditions. They also need to document better the reduction in ATM levels in response to ATM siRNA (Figure 1—figure supplement 1B is not convincing in this regard).

2) Some of the SIRT6 localization experiments (Figure 2C, top middle panel) are not entirely convincing. In general, these experiments are quite confusingly presented and hard to follow, as the authors keep switching colors (SIRT6 is sometimes green, and sometimes red), and not every panel is labeled.

3) In Figure 5E, why does LacRSIRT6 sometimes localize to a single point in the cell (top left), and sometimes show mostly diffuse localization (top right).

4) If the authors want to substantiate the interaction of SIRT6 with Mre11 and NBS (Figure 5E), they need to assess interaction of the endogenous proteins, not just overexpressed tagged transgenes.

5) It would be helpful for the authors to show in detail how they believe DNA interacts with the tunnel on SIRT6 in their model. In particular, two of the apparently key SIRT6 residues are aspartates, which are negatively charged and would repel the negatively charged backbone on DNA.

6) I don't think the NAM experiment (S5A-B) is sufficient to rule out catalytic SIRT6 roles, particularly since the authors don't actually directly test SIRT6 activity in response to NAM treatment. A much cleaner system would be to reconstitute SIRT6 KO cells with WT or appropriate site-directed mutants of SIRT6.

7) Likewise, the manuscript would be much stronger if the authors showed that some of their DNA-binding mutants maintained a "canonical" function of SIRT6 in vivo (e.g. acH3K9 or acH3K56 acetylation), but failed to rescue DNA damage resistance of SIRT6 KO cells in vivo.

*Reviewer #2:*

In their manuscript entitled "SIRT6 is a DNA Double-Strand Break Sensor," Onn and colleagues propose that SIRT6 plays a previously uncharacterized role as a DNA damage sensor, which is critical for initiating the DNA damage response (DDR). They further report that other sirtuins share DSB binding capacity and DDR activation. Overall, the authors suggest that SIRT6 functions very proximal in the DDR, prior to repair pathway selection. Their findings are potentially relevant and support previous reports on the involvement of SIRT6 in the DDR. However, we felt that some of their findings, as outlined below, were not sufficiently supported by their data or that more information needs to be included with their current report before it can be recommended for publication.

Substantive concerns

In Figure 1, formation of large SIRT6-GFP foci in response to laser-mediated UV irradiation is shown. No information on timing is given. Moreover, it is not directly shown that these Sirt6 foci correspond to DSB foci (e.g., via gH2AX or 53BP1 co-staining). It is not discussed what fraction of cells actually show accumulation of SIRT6 in foci post UV irradiation. Under these laser irradiation conditions, how many DSB foci per cell are generated?

A major conclusion of this manuscript is that SIRT6 DSB recruitment is independent of DDR factors. These conclusions are based on knockdown or chemical inhibition of factors. The ATM depletion data shown in Figure 1—figure supplement 1B is not compelling. Additional experiments showing inhibition of ATM function are necessary to confirm DDR-independent recruitment of SIRT6 to DSBs. Similarly, it should be confirmed that RNF8-dependent processes are indeed inactivated under the experimental conditions of RNF8 knockdown. It should further be stated how many cells were analyzed per experiment shown in Figure 1A and B.

*Reviewer #3:*

This study proposes an intriguing model that SIRT6 directly binds DNA breaks and acts at the very top of the DDR cascade. In principle, aspects of this model would improve the granularity of how we understand SIRT6 function in DNA repair processes. However, previous studies, including from the corresponding author, already proposed a very early function of SIRT6 in DDR, and the present manuscript falls short of providing more than an incremental advance. There are also considerable concerns that some important controls are lacking, and some data do not strongly support the conclusions and/or are overinterpreted. Thus, overall, the work is not appropriate for *eLife*.

1) A key conclusion of the study is that SIRT6 is recruited to DNA damage sites independent of known DDR factors ATM, H2AX, RNF8, and Parp. In the case of Parp, the data are clear; and MacromKate2 is an appropriate positive control for Parp inhibition. By contrast, in the cases of ATM, H2AX, and RNF8, it is not clear that these factors are functionally inhibited. The immunoblots (Figure 1—figure supplement 1) are not convincing that ATM or RNF8 are adequately depleted, and some functional demonstration that signaling by these factors is abrogated is necessary to conclude that SIRT6 recruitment to DSBs is indeed independent of the factors. Regarding Parp, previous work has shown that SIRT6 is upstream of Parp (Mao et al., 2011), so the finding is not surprising.

2) In Figure 2 and Figure 2—figure supplement 1, the authors claim that in tethering assays, SIRT6 colocalizes with multiple DDR factors (SNF2H, MRE11, Ku80, NBS1) but not ATM. It is very difficult to conclude this from the data shown. It is not at all clear how the co-localization was scored, since the SIRT6 signal is largely diffuse, and there certainly does appear to be yellow (signal overlap) in the ATM-SIRT6 image. More importantly, the tethering assay may be overinterpreted if used to make conclusions about whether one factor is upstream of another; instead, it may simply be a readout for protein interactions. Indeed, in Figure 5, reciprocal assays tethering SIRT6 are used to claim that SIRT6 initiates signaling of downstream factors. By the same logic, then the data in Figure 2 would indicate that SIRT6 is recruited by MRE11/NBS1/Ku80 etc. Such interpretations are problematic.

3) The notion that ssDNA binds SIRT6 in a tunnel-like structure is intriguing. However, it is surprising that mutation of key residues have relatively minor effects on DNA binding in vitro (Figure 4E). Moreover, mutation in the N-terminus, which in Figure 4D is shown not to be necessary for binding, has just as much effect as the putative tunnel forming residues.

4) In Figure 5 and Figure 5—figure supplement 1, it is troubling that the diffuse signal of LacR-SIRT6 varies greatly in different panels. In some the entire nucleus has strong signal, whereas in others, the nucleus has virtually no signal except for the lacR-recruited dot. What is the basis for this, and it would seem to make the data problematic.

[Editors’ note: further revisions were suggested after the authors resubmitted, as described below.]

Thank you for submitting your article "SIRT6 is a DNA double-strand break sensor" for consideration by *eLife*. Your article has been reviewed by three peer reviewers, one of whom is a member of our Board of Reviewing Editors, and the evaluation has been overseen by Jessica Tyler as the Senior Editor. The reviewers have opted to remain anonymous.

The reviewers have discussed the reviews with one another and the Reviewing Editor has drafted this decision to help you prepare a revised submission.

Summary:

This manuscript by Onn and colleagues proposes that the mammalian sirtuin protein SIRT6 directly binds to damaged DNA to initiate DNA damage responses. These findings, if fully substantiated, would add to the molecular understanding of SIRT6 function, and make important contributions to sirtuin biology and the study of DNA damage repair.

At present, however, the manuscript falls short of providing a sufficiently clear picture of the mechanism(s) and physiological relevance of SIRT6 as a DSB sensor. It is viewed that several major issues must be addressed in order for the conclusions to be rigorously convincing. There is concern over the heavy reliance on the LacO-based "artificial" system to follow the role of SIRT6 as a DNA break sensor and the rigor of the experimental interpretations generally. There is also a tendency in the manuscript to over interpret/ overstate conclusions, and a careful editing will be required to avoid this. We have the following requested revisions.

Essential revisions:

1) The authors use the PARP inhibitor Olaparib to show that SIRT6 binds DSBs independent of PARP1. However, these experiments do not address if PARP protein (independent of activity) might recruit SIRT6; to address this, the authors need to deplete PARP1 by RNAi or CRISPR knockout, and test if this affects SIRT6. Further, the authors state that SIRT6 binding to DSBs is independent of other proteins. Yet, at least in vivo this could be mediated by PARP1 or other DNA binding proteins that were not examined (potentially WRN).

2) The authors base their model heavily on the LacO tethering system, but there remains major concern that results using this system are insufficient to justify their conclusions regarding DNA damage signaling and protein interactions of SIRT6. For example, to make any statements about SIRT6 interacting with other DDR factors, the LacO system is insufficient, as acknowledged by the authors. The authors failed to perform co-IP experiments, which is a concern. Did they try to IP from both directions? Also after damage? If not, care must be taken not to overstate their conclusions from the tethering assay.

3) The authors have not adequately addressed the prior concerns over the scoring of "co-localization" in the tethering assays. As pointed out before, in Figure 2B, for example, the graph shows ~55% "recruitment" of SIRT6 by SNF2H-LacR but only background levels by ATM-LacR. However, the primary data in 2C don't reflect this at all. Indeed, there seems to be clear overlap (yellow dot) in the ATM samples, at least as clear if not clearer than for the SNF2H tethering. It's not at all obvious how such data could lead to the graph, and suggests the scoring is highly subjective. The authors' response merely describes the scoring process, but does not deal with this discrepancy.

4) There remains concern over the basis on which the authors draw conclusions about what factor recruits what, and this affects the main conclusion that SIRT6 is initiating at the top of the DSB response. In the response, they say their data "suggest that SIRT6 arrives independently and can recruit these factors, or it can be recruited by them when they initiate the signal." This suggests, as was raised in previous review, that the tethering assays that are employed are insufficient to draw conclusions about which factor is upstream of another. More important, the point by the authors undercuts the strength of their model that SIRT6 is at the very top.

5) Figure 3A-C. the authors use EMSA to calculate the Kd value and cooperativity for SIRT6-DNA binding. This method is semi quantitative and therefore additional quantitative methods should be used such as Surface Plasmon Resonance (SPR) and Biolayer interferometry (BLI). Moreover, given the fast interaction of SIRT6 with the damaged site (5 seconds), a more reliable Kd value based on the methods listed above can provide additional information for binding kinetics (k_on_ and K_off_) which is needed in order to justify their claims.

6) Several SIRT6 ChIP seq experiments were published, which involves sonicating DNA and the creation of DNA breaks. If SIRT6 binds to ssDNA as the authors propose, it is surprising that ChIP-seq studies were able to identify clear peaks in vivo. This suggests that the ssDNA binding may not occur physiologically in vivo.

7) For reasons outlined in the rebuttal letter, prior data in Figure 1 is no longer included. The new data are less compelling. Overall effects are very moderate in size. Also, in Figure 1B, it is not clear what n = 10 refers to. Is it cells or perhaps independent experiments? Error bars are not included, and p-values are not shown for Figure 1 data. This is of concern and needs to be addressed. We noted that p-values are stated for other data (but not Figure 1 data) in the Transparent Reporting Form.

8) The authors state in their rebuttal that the U2OS cells used were "very unstable" and that "too many gH2AX foci were present making the experiments impossible to quantify or rely on. When cells were too old or damaged, we thawed new ones from old passages and began the whole experiment again". This seems quite arbitrary and made us concerned about the overall validity and rigor of the studies. At a minimum, more specific criteria for what constitutes cells being "too old or damaged" would need to be clearly described in the manuscript.

9) The paper needs much better editing. The authors indicate that the revised manuscript was sent for "editorial revision" but similar issues as pointed out in the prior reviews remain.

Aside from typos, there are numerous mistakes in the reference list. For example, Tang is mentioned 3 times in a row, Kaidi et al. was retracted, etc.

[Editors' note: further revisions were suggested prior to acceptance, as described below.]

Thank you for resubmitting your work entitled "SIRT6 is a DNA double-strand break sensor" for further consideration by *eLife*. Your revised article has been evaluated by Jessica Tyler as the Senior Editor, and a Reviewing Editor.

The manuscript has been improved but there are some remaining issues that need to be addressed before acceptance, as outlined below:

1) The revised manuscript now describes more accurately that Parp activity (versus Parp1 protein) is dispensable for SIRT6 binding to DSBs; however, it is problematic that the new submission still fails to show that Parp1 protein does not recruit SIRT6 to breaks. Although knocking-down or CRISPR KO of Parp1 may require trouble shooting for technical reasons, it remains concerning that this essential experiment has not been done, because this limits the conclusions that can then be drawn. The authors' state that Mao et al. previously showed that Parp1 KO MEFs have no defects in SIRT6 recruitment to DSBs; however, this appears incorrect. In fact, Mao et al. demonstrated that SIRT6 and Parp1 interact physically with each other. This makes it a distinct possibility that the SIRT6-Parp1 interaction could in fact mediate recruitment of SIRT6 to DSBs, as previously raised by the reviews (though this is not the focus of the Mao paper). Because this possibility has not been ruled out, the authors cannot rigorously draw the conclusion that SIRT6 binds DSBs "independent of known DSB sensors". For publication in *eLife*, the authors need to be very careful to limit their conclusions to what their data actually show. For example, they need to rephrase their conclusions (throughout the text), to "SIRT6 recruitment is independent of the MRE11 and Ku DSB sensors and of Parp1 enzymatic activity," or similar statement. If they wish to propose the model that SIRT6 is independent of all known sensors, they will need to state explicitly that this is a speculation, and that their data cannot rule out that recruitment of SIRT6 to DSBs might occur at least in part via its previously described interaction with Parp1.

2) There is also remaining concern that conclusions continue to be overstated in places. In the last sentence of the Discussion ("In conclusion, we have demonstrated that Sirtuins – and mainly SIRT6 – have a role as independent DNA damage sensors."), the conclusion regarding all Sirtuins is not supported by the data. The authors should limit their conclusion to SIRT6. There is not sufficient evidence from their experiments that the other Sirtuins have roles as sensors.

---

## [Author Response]

[Editors’ note: the authors resubmitted a revised version of the paper for consideration. What follows is the authors’ response to the first round of review.]

Reviewer #1:[…] 1) For Olaparib studies, the authors need to show that PAR levels are diminished under their treatment conditions. They also need to document better the reduction in ATM levels in response to ATM siRNA (Figure 1—figure supplement 1B is not convincing in this regard).

MacroH2A is commonly used as a control for Olaparib inhibition (Sherry et al., 2017), as reviewer #3 noted. Given that we cannot perform immunofluorescence at such early time points (these experiments are in live imaging), we show lack of Parylation in live imaging through the lack of MacroH2A recruitment.

Regarding the shATM blot, we replaced the previous blot with a new one with better visible difference in ATM levels. It was achieved by using a different and better antibody (Figure 1—figure supplement 1H).

To further prove the independence of SIRT6 on other DDR factors, we silenced MRE11 and Ku80 which are the two main DSB sensors (in addition to PARP proteins). As expected, we observed defective recruitment of the main proteins in their complexes- NBS1 (for MRE11) and Ku70 (for Ku80). However, the silencing of neither MRE11 nor Ku80 affected SIRT6 recruitment. (Figure 1C-D, Figure 1—figure supplement 1C-F).

Interestingly, when we tested the effect of SIRT6-KO on the recruitment of MRE11 and Ku80, we found that while the recruitment of MRE11 was defective, Ku80 was not affected by the lack of SIRT6, suggesting a more prominent role of SIRT6 in HR. (Figure 1—figure supplement 1G, 1E-H).

2) Some of the SIRT6 localization experiments (Figure 2C, top middle panel) are not entirely convincing. In general, these experiments are quite confusingly presented and hard to follow, as the authors keep switching colors (SIRT6 is sometimes green, and sometimes red), and not every panel is labeled.

We have improved this issue by adding a label to each image so as to avoid confusions. Since in this work we used a large verity of plasmids and proteins, we were not always able to obtain every plasmid in every color. To overcome this issue, we obtained both SIRT6-GFP and SIRT6-RFP and chose which one to use according to the color of the other protein in each experiment. For the same reason, we also generated both SIRT6-LacR-GFP and SIRT6-LacR-RFP, as well as GFP-LacR and Cherry-LacR.

It is important to note that these experiments were manually quantified by two different students (several of the experiments were quantified by a second student in a blind manner to assure a proper quantification process).

Since the amount of noise in the assay could be quite large, we set a number of rules on what to quantify and how to do it:

Quantified cells were chosen to obtain the following characteristics:

1) Cells must be co-transfected (meaning, they express both red and green colors to avoid lack of co-localization due to the lack of one of the proteins from the cells)

2) Strong signaling and large Lac-Protein dot (that was clearly not background, or the protein was not localized at the LacO repeats).

3) No cross channel fluorescence (we observed minimal to zero crossover and set a threshold to ignore this low false signal)

4) U20S were very unstable and several times too many ϫH2AX foci were present making the experiments impossible to quantify or rely on. When cells were too old or damaged, we thawed new ones from old passages and began the whole experiment again.

3) In Figure 5E, why does LacRSIRT6 sometimes localize to a single point in the cell (top left), and sometimes show mostly diffuse localization (top right).

SIRT6 is a nuclear/chromatin bound protein spread in the nucleus. We usually see the entire nucleus in green as in the sample of the diffused localization. In some pictures, the LacR dot is very bright and clear, and little background is observed. In those cases, the pictures were taken with lower exposure times, and there appears to be no SIRT6 in the nucleus, though in the background there was. In some pictures with lower expression or higher background, we used a longer time of exposure. The causes of these differences depend on: the levels of expression, the time after transfection and the position in the well (well borders tend to have higher background intensity). This does not represent a difference in SIRT6 behavior (as some SIRT6-GFP can be seen in the second top left corner (new Figure 6B).

Importantly, all of the pictures were taken before saturation and avoiding channels cross-talk since, as explained before, these experiments were quantified in a qualitative manner and the fluorescence intensity was not relevant for the co-localization. We allowed ourselves to take different exposure times following the rules detailed before.

It is also important to note that SIRT6 does not form foci and even after irradiation SIRT6 still appeared diffused, covering the entire nucleus. To prove it localizes to the sites of damage, we previously used other methods such as laser-induced damage in live imaging, as shown in this manuscript. But we also showed it by laser induced damage and IF, and ChIP on sites cleaved by ISceI (tested before in (Toiber et al., 2013)). Thus, an apparently large SIRT6 foci in this system is due to the accumulation of a large number of LacR-SIRT6 proteins in the LacO site 256 repeats.

4) If the authors want to substantiate the interaction of SIRT6 with Mre11 and NBS (Figure 5E), they need to assess interaction of the endogenous proteins, not just overexpressed tagged transgenes.

After several attempts, we failed to Co-IP endogenous SIRT6 with endogenous MRE11 and NBS1 due to excessive background with various antibodies, therefore, we removed the Flag-IP. However, we also took a different approach to distinguish between interaction, and recruitment dependent on signaling. We performed the tethering assay while inhibiting DDR signaling using wortmannin- an ATM, ATR and DNA-PKc inhibitor. This treatment led to a reduction in H2AX phosphorylation and in co-localization of SIRT6-LacR with both 53BP1 and BRCA1. In this experiment, we did not observe the same reduction in co-localization with MRE11 and Ku80. The most reasonable explanation for these results is that the recruitment of these factors is based on direct protein-protein interactions or complex formation, and not on DDR signaling. However, we cannot confirm this is a protein-protein interaction, therefore, we removed the IP and the paragraph from this work, and added only the wortmannin experiment. (Figure 6—figure supplement 3A-E).

5) It would be helpful for the authors to show in detail how they believe DNA interacts with the tunnel on SIRT6 in their model. In particular, two of the apparently key SIRT6 residues are aspartates, which are negatively charged and would repel the negatively charged backbone on DNA.

We propose the following spatial model of SIRT6. Amino acids predicted to bind to DNA are colored in red. The tunnel is located at the bottom part, with a single DNA strand drawn inside (Figure 4E).

As for the negatively charged amino acids, aspartate could be involved in binding the bases and backbone (binding of Asp to Cys). Usually, Asp is part of a sequence that binds DNA through the divalent metal ion (Mg (II)), and mediates the binding to the phosphate backbone of nucleic acids (Akabayov et al., 2011). At this point, and without a crystal structure, it is impossible to fully determine how the binding is performed. We do believe, however, that the binding is influenced by hydrophobic interactions. We would like to point out that our structural model should be regarded as a hypothetical model, which we have tried to make clear in the text.

**Author response image 1. respfig1:** Aspartate basepairing with DNA bases and backbone.

6) I don't think the NAM experiment (S5A-B) is sufficient to rule out catalytic SIRT6 roles, particularly since the authors don't actually directly test SIRT6 activity in response to NAM treatment. A much cleaner system would be to reconstitute SIRT6 KO cells with WT or appropriate site-directed mutants of SIRT6.

The U20S cellular system developed by Roger Greenberg’s lab is resistant to various antibiotics, thus the knockout of SIRT6 in these cells failed. However, taking into account our findings, we believe that this action is not necessary. To prove that SIRT6 catalytic activity is not important for DDR initiation, we used the catalytic mutant SIRT6 H133Y, which is known to be catalytically dead. This was confirmed by a fluor de lys assay performed in this work as well (Figure 4—figure supplement 1C).

To overrule any catalytic activity performed by possible dimerization of SIRT6- HY-LacR with the endogenous SIRT6 present in the cells, we inhibited its activity using Nicotinamide (NAM).

We have now added the control showing an increase in the acetylation of the SIRT6 target H3K56ac, indicating that NAM treatment for 12 and 24 hours inactivates Sirtuins activity. Again, no change was found in SIRT6-HY-LacR ability to initiate the DDR, suggesting that this ability is not dependent on SIRT6 catalytic activity (Figure 5—figure supplement 1A-B). The previous experiment was replaced by this new figure.

7) Likewise, the manuscript would be much stronger if the authors showed that some of their DNA-binding mutants maintained a "canonical" function of SIRT6 in vivo (e.g. acH3K9 or acH3K56 acetylation), but failed to rescue DNA damage resistance of SIRT6 KO cells in vivo.

It would be most interesting to find separate functions, however, most of the mutants affect the catalytic activity of SIRT6 (Figure 4—figure supplement 1C) as well as its binding activity. We think that SIRT6 binding domain to DNA is very close to the binding of NAD, since both NAD and ssDNA have a very similar chemical composition. We hypothesize that this is the reason why most of the mutations in the tunnels also affect the catalytic activity of SIRT6. However, we do observe two exceptions: A13W and D63Y. On the one hand, D63Y mutation had no effect on DNA binding and it causes a significant reduction in SIRT6 catalytic activity. A13W mutation, on the other hand, resulted in an increase in catalytic activity along with a reduction in DNA binding. We raise this point in the discussion of the paper as well. However, with the lack of a definitive proof (such as crystal structure) we believe that making any unequivocal conclusions on whether these two abilities are related to one another or not is out of the scope of this paper.

Reviewer #2:Substantive concernsIn Figure 1, formation of large SIRT6-GFP foci in response to laser-mediated UV irradiation is shown. No information on timing is given. Moreover, it is not directly shown that these Sirt6 foci correspond to DSB foci (e.g., via gH2AX or 53BP1 co-staining). It is not discussed what fraction of cells actually show accumulation of SIRT6 in foci post UV irradiation. Under these laser irradiation conditions, how many DSB foci per cell are generated?

In these experiments, laser induced damage is used in a live imaging technique which generates DNA damage at a single (see scheme) relatively large dot on cells pre-sensitized with BrdU or Hoechst, thus generating double strand breaks. Cells are transfected with a florescent protein plasmid, and the recruitment of this protein to sites of damage is measured in vivo. Images were taken at 3-5 second intervals for 60 seconds after photobleacing. Before bleaching, 3 images were taken for each cell to measure the florescence baseline, see Author response image 2.

**Author response image 2. respfig2:** Scheme of Laser Induced damage (LID) experiment.

In this method we do not generate DNA damage foci. Therefore, there are no quantifications of the percentage of cells that show foci, but we show an analysis on a single cell that was measured over time. The experiment was repeated in several cells (note the N = in each experiment) and the results are presented as the average of the measurements.

The method was explained in (Toiber et al., 2013) and in here for clarification. It was adapted to microscope use for the following experiments:

“Fluorescence Recovery after Photobleaching (FRAP)

FRAP experiments (Laser induced damage) were performed as previously described at Toiber et al., 2013. […] Images analysis was performed using ImageJ 1.52i software.”

In Figure 1A-B and Figure 1—figure supplement 1A-B, U2OS Cells were incubated with 10µM BrdU or 10µM BrdU + 10µM Olaparib over-night, before going through the same procedure.

A major conclusion of this manuscript is that SIRT6 DSB recruitment is independent of DDR factors. These conclusions are based on knockdown or chemical inhibition of factors. The ATM depletion data shown in Figure 1—figure supplement 1B is not compelling. Additional experiments showing inhibition of ATM function are necessary to confirm DDR-independent recruitment of SIRT6 to DSBs. Similarly, it should be confirmed that RNF8-dependent processes are indeed inactivated under the experimental conditions of RNF8 knockdown.

See response to reviewer #1.

It should further be stated how many cells were analyzed per experiment shown in Figure 1A and B.

This information was added to the manuscriptoiber51636Authorresponseimage2.epsoiber51636Authorresponseimage2.eps

Reviewer #3:1) A key conclusion of the study is that SIRT6 is recruited to DNA damage sites independent of known DDR factors ATM, H2AX, RNF8, and Parp. In the case of Parp, the data are clear; and MacromKate2 is an appropriate positive control for Parp inhibition. By contrast, in the cases of ATM, H2AX, and RNF8, it is not clear that these factors are functionally inhibited. The immunoblots (Figure 1—figure supplement 1) are not convincing that ATM or RNF8 are adequately depleted, and some functional demonstration that signaling by these factors is abrogated is necessary to conclude that SIRT6 recruitment to DSBs is indeed independent of the factors. Regarding Parp, previous work has shown that SIRT6 is upstream of Parp (Mao et al., 2011), so the finding is not surprising.

Regarding the first concern, please see response to reviewer #1.

Regarding the relationship between SIRT6 and PARP, Mao and associates (Mao et al., 2011) indeed deal with this question in their work; however, they mainly focus on the relationship under oxidative stress induced by H2O2. Moreover, they do not address the question of which protein arrives to sites of damage first. PARP1 is the fastest known enzyme to be recruited to sites of DSBs (Yang et al., 2018), and is responsible, at least in part, for the recruitment of the MRN complex (Haince et al., 2008). Taking that into consideration, along with the fact that SIRT6 was not reported to be a DSB binding protein before, its arrival to sites of damage independently of PARP1, even if not surprising, is a significant finding.

2) In Figure 2 and Figure 2—figure supplement 1, the authors claim that in tethering assays, SIRT6 colocalizes with multiple DDR factors (SNF2H, MRE11, Ku80, NBS1) but not ATM. It is very difficult to conclude this from the data shown. It is not at all clear how the co-localization was scored, since the SIRT6 signal is largely diffuse, and there certainly does appear to be yellow (signal overlap) in the ATM-SIRT6 image. More importantly, the tethering assay may be overinterpreted if used to make conclusions about whether one factor is upstream of another; instead, it may simply be a readout for protein interactions. Indeed, in Figure 5, reciprocal assays tethering SIRT6 are used to claim that SIRT6 initiates signaling of downstream factors. By the same logic, then the data in Figure 2 would indicate that SIRT6 is recruited by MRE11/NBS1/Ku80 etc. Such interpretations are problematic.

Reviewer #3 is correct to point out that in the tethering assay the data can be interpreted as protein-protein interactions (direct interaction: protein-protein; or recruitment: signaling induced co-localization, but not through direct interaction). However, the early arrival of SIRT6, together with the new results of MRE11 and Ku80 silencing, allow us to suggest that SIRT6 arrives independently and can recruit these factors, or it can be recruited by them when they initiate the signal. This would allow SIRT6 to be a sensor, but also to form protein-complexes with other sensors, revealing the importance for both HR and NHEJ.

A better explanation on how co-localization was scored can be found in the response to reviewer #1. However, it is important to note that SIRT6 tends to be evenly dispersed in the nucleus and does not form foci, thus, a visible SIRT6 foci in the tethering assay can only be explained by the accumulation of a large number of SIRT6 molecules at the LacO site. Since there is no actual DNA damage in this site (as shown by GFP-LacR control), we can assume that the accumulation of DDR proteins is initiated by the system.

To better understand which co-localizations occur in response to DDR signaling and which occur due to protein interactions, we performed the tethering assay inhibiting DDR signaling with wortmannin- an ATM, ATR and DNA-PKc inhibitor. While signaling was inhibited (seen by a reduction in co-localization with ϫH2AX), as well as the recruitment of both 53BP1 and BRCA1, the recruitment of MRE11 and Ku80 was unaffected (Figure 6—figure supplement 3A-E). These findings suggest that while there is a direct interaction between SIRT6 and MRE11 or Ku80, the most downstream factors arrive upon signaling.

3) The notion that ssDNA binds SIRT6 in a tunnel-like structure is intriguing. However, it is surprising that mutation of key residues have relatively minor effects on DNA binding in vitro (Figure 4E).

These experiments were carried out with a single point mutant to avoid instability or the complete misfolding of the protein. Since these are just single point mutations, we do not expect that they would completely abolish SIRT6 DNA binding ability, but rather reduce it with an almost 50% reduction for SIRT6-HY mutant.

Moreover, mutation in the N-terminus, which in Figure 4D is shown not to be necessary for binding, has just as much effect as the putative tunnel forming residues.

Figure 4D shows the core domain (not the N terminus), where most of the residues predicted to interact with DNA are present, as well as the ones we mutated (Figure 4A-C, S4A).

4) In Figure 5 and Figure 5—figure supplement 1, it is troubling that the diffuse signal of LacR-SIRT6 varies greatly in different panels. In some the entire nucleus has strong signal, whereas in others, the nucleus has virtually no signal except for the lacR-recruited dot. What is the basis for this, and it would seem to make the data problematic.

See response to reviewer #1.

[Editors’ note: what follows is the authors’ response to the second round of review.]

Essential revisions:1) The authors use the PARP inhibitor Olaparib to show that SIRT6 binds DSBs independent of PARP1. However, these experiments do not address if PARP protein (independent of activity) might recruit SIRT6; to address this, the authors need to deplete PARP1 by RNAi or CRISPR knockout, and test if this affects SIRT6. Further, the authors state that SIRT6 binding to DSBs is independent of other proteins. Yet, at least in vivo this could be mediated by PARP1 or other DNA binding proteins that were not examined (potentially WRN).

1a) We agree with the reviewer that we cannot rule out direct interaction of PARP1 with SIRT6. To clarify this point, we rephrase the interpretation of our results (PARP activity).

We tried to generate PAPR1-KO cells, but were unsuccessful (see Author response image 3). However, Mao et al^1^ show no effect in SIRT6 recruitment in MEFs KO for PARP1.

We would like to emphasize, however, that this lack of SIRT6 dependence on PARP1 activity is contrasted by the fact that MRE11 and NBS1 recruitment to DSBs does require PARP1 activity^2^. On a technical note, Olaparib inhibits all PARPs (PARP1, PARP2 and PARP3), which could not be accomplished by KO of PARP1 alone.

**Author response image 3. respfig3:** Western blot of Hela cells after attempt to create PARP1 KO using a CRISPR-Cas9 system.

1b) When addressing SIRT6 recruitment to sites of DSBs and its DNA binding ability, it is important to take into consideration the timing of protein recruitment to sites of damage.

The most relevant proteins to test are the first proteins to arrive to DSB sites. These are the other DSB sensors, which also initiate the DNA damage response. If SIRT6 arrival is independent of these proteins (as indeed verified as shown in figure 1 and Figure 1—figure supplement 1), then it is not likely that SIRT6 would be dependent on proteins that arrive at the site of damage at later time points. It is an interesting question whether WRN and other proteins influence SIRT6 repair efficiency, kinetics and other parameters of repair; however, these are beyond the scope of this paper.

Therefore, we suggest that with the elimination of the known sensors (PAPR inhibition, MRE11 and KU80), SIRT6 arrival should be independent of the activity of the hundreds of other proteins that participate downstream in the DNA damage response.

2) The authors base their model heavily on the LacO tethering system, but there remains major concern that results using this system are insufficient to justify their conclusions regarding DNA damage signaling and protein interactions of SIRT6. For example, to make any statements about SIRT6 interacting with other DDR factors, the LacO system is insufficient, as acknowledged by the authors. The authors failed to perform co-IP experiments, which is a concern. Did they try to IP from both directions? Also after damage? If not, care must be taken not to overstate their conclusions from the tethering assay.

We agree with the reviewer that our data do not resolve the possible contribution of direct and indirect protein interactions to SIRT6 DNA damage activity. We have indeed attempted several versions of the endogenous Co-IP from both directions as well as post irradiation, but these attempts were unsuccessful because of the poor quality of the antibodies.

However, we used a different approach to differentiate between recruitment triggered by protein-protein interactions and signaling-based recruitment. We performed the LacO tethering assays using the ATM/ATR/DNA-PK inhibitor Wortmannin. We show that addition of Wortmannin inhibited DDR signaling (manifested in reduction of ɤH2AX levels) and the recruitment of downstream factors, such as 53BP1 and BRCA1. However, the inhibition by Wortmannin had no effect on the recruitment of MRE11 and Ku80, suggesting that their recruitment occurs independently of the signaling propagated by ATM/ATR/DNA-PK kinases. As far as we know, any protein recruitment that occurs prior to this stage is based on direct interaction or complex formation, supported also by the reciprocal recruitment of SIRT6 and these factors.

We therefore suggest in the manuscript that protein interactions could take part in the SIRT6 signaling/complex formation (Figure 6—figure supplement 3), without definitively claiming that such interactions exist. In any case, we do not think that this diminishes the importance of our findings and conclusions, since the capacity of SIRT6 to initiate recruitment remains relevant regardless of the contribution of protein interactions.

We would like to highlight that one of the most important finding of our work is not whether the interaction is direct or indirect, but rather that the recruitment of SIRT6 is not triggered by the DDR signaling (e.g., via ATM). This, in turn, suggests that SIRT6 does not function downstream to ATM. Signaling itself is not sufficient to recruit SIRT6 in the absence of open-ended DNA. This, together with our biochemical assays, indicates that SIRT6 directly binds open-ended DNA, suggesting that the mechanism is the direct binding of DSBs by SIRT6.

We disagree that we base our model heavily on the LacO system, since many of our findings are based on other several assays (laser induced damage, EMSA, DNA binding assay, SAXS, end protection assay, etc.). Moreover, taking into account previous publications indicating that silencing of SIRT6 completely impaired the recruitment of BRCA1, RPA, 53BP1, SNF2H, H2AUb, and the ability to repair DNA damage itself through both HR and NHEJ, strengthens our point of SIRT6 as a DSB sensor.

3) The authors have not adequately addressed the prior concerns over the scoring of "co-localization" in the tethering assays. As pointed out before, in Figure 2B, for example, the graph shows ~55% "recruitment" of SIRT6 by SNF2H-LacR but only background levels by ATM-LacR. However, the primary data in 2C don't reflect this at all. Indeed, there seems to be clear overlap (yellow dot) in the ATM samples, at least as clear if not clearer than for the SNF2H tethering. It's not at all obvious how such data could lead to the graph, and suggests the scoring is highly subjective. The authors' response merely describes the scoring process, but does not deal with this discrepancy.

The tethering assay was developed in the lab of Tom Misteli and published in Science in 2008^3^. Ever since, it has been widely used by his lab and others to show causality in DNA damage response initiation and protein recruitment^4–8^.

In order to alleviate the doubts of the reviewers, it is important to note that if two fluorescent proteins (GFP and RFP) co-exist in the nucleus in similar intensities, the cell will most likely present a “yellow dot” every time a merged image is presented. However, this is not considered as a co-localization. In our scoring process that will be exemplified below, co-localization is only positive if in the exact position of the LacR protein there is a clear “dot” in the other channel.

To better explain the method, we have added here a more detailed explanation of the experimental procedure, data collection and the analysis process that is now part of the Materials and methods.

“Tethering assay

U2OS cells containing 256X LacO sequence repeats in their genome were transfected with plasmids of chimeric LacR-DDR enzyme-GFP/Cherry proteins. Cells were either co-transfected with a second plasmid of a fluorescent/ Flag tagged protein or Immuno-stained (See Immunofluorescence) for an endogenic protein.”

Out of the transfected cells, those chosen for the experiment had to adhere to the following conditions:

1) Cells exhibiting a large visible focus of LacR-DDR-GFP/Cherry at LacO sites (located using fluorescent microscopy).

2) Cells must express both proteins of interest, meaning they should exhibit both red and green colors.

3) No cross channel fluorescence (we observed minimal to zero crossover and set a threshold to ignore this low false signal).

After cell selection, co-localization percentage was assessed. Co-localization is defined as the common localization of large foci of the two proteins of interest at the LacO site. Nuclear proteins often seem to be evenly diffused throughout the entire nucleus; thus, unless 2 defined foci have been located, this is not defined as co-localization.

However, to better explain the scoring process and the different variations that could be seen in these experiments, we performed a computational analysis for numerous cells, using ImageJ 1.52p software (see examples in Author response image 4). In the analysis, we:

1) Draw a line crossing the nucleus at the focus formed by the LacR- chimeric protein.

2) Measure the fluorescence intensity at every pixel along the line for both filters.

**Author response image 4. respfig4:** Co-localization analysis. (**A**) LacR-ATM with SIRT6-GFP: Both are nuclear proteins that are usually found evenly spread across the entire nucleus. While we can see a sharp peak where the ATM-LacR focus is located, we do not see a peak of SIRT6-GFP at the same cellular location, which means there is no co-localization of the two proteins in this image. (**B**) LacR-SNF2H with SIRT6-RFP: A clear peak in the graph of the SNF2H-LacR is accompanied by a clear peak of the SIRT6-RFP, meaning positive co-localization. Since the red signal is much weaker than the green, the SIRT6 peak is much smaller than the SNF2H one. (**C**) LacR-Ku80 with SIRT6-RFP: In this case the SIRT6-RFP signal is much stronger that the Ku80-LacR GFP signal; however, we can still observe clear peaks at the same location. (**D**) LacR-GFP with SIRT6-RFP: In this negative control condition, there are no parallel peaks in the graph. However, when looking at the merged image we see that both cells are”yellow”. This just means that both signals are equally as strong (as can be seen in the graph as well) and that both proteins are spread throughout the entire nucleus; however, this is not a co-localization. (E-F) LacR-SIRT6 with 53BP1: When comparing endogenous 53BP1 (**E**) to the exogenous (**F**), we can see that using an endogenous antibody usually gives a “cleaner” higher peak with less background; however, in both cases the co-localization is clear.

Colocalization quantification:

Co-localization was assessed as either positive (1) or negative (0). From this analysis, the percentage of cells that exhibit co-localization (positive cells) was calculated, and defined as “percentage of co-localization between two proteins”. The graphs that appear in the paper present the average of several individual experiments with SEM, while “n” is the number of cells used.

We are aware of a concern recently brought up concerning this method ^9^, in which it was claimed that the LacO array is susceptible to DNA damage. However, we address this issue by including a negative control of GFP/mCherry-LacR protein in all of our experiments.

Lastly, to avoid any further confusion that might arise from the images selected for Figure 2 of the paper, we replaced the ATM-LacR image with a different one, hopefully making our claim clearer.

After performing hundreds of these experiments over many years we are highly trained at assessing co-localization; thus we feel comfortable about quantifying the experiments manually. Moreover, many of the experiments were quantified blindly by more than one student in order to reduce subjectivity and assure a proper quantification process.

4) There remains concern over the basis on which the authors draw conclusions about what factor recruits what, and this affects the main conclusion that SIRT6 is initiating at the top of the DSB response. In the response, they say their data "suggest that SIRT6 arrives independently and can recruit these factors, or it can be recruited by them when they initiate the signal." This suggests, as was raised in previous review, that the tethering assays that are employed are insufficient to draw conclusions about which factor is upstream of another. More important, the point by the authors undercuts the strength of their model that SIRT6 is at the very top.

There seem to be some misunderstanding about the suggested model. Just like MRE11 and Ku80 arrive to sites of damage independently of each other^11,12^, we suggest that SIRT6 can arrive independently of them as well. There is no factor at the top of all DSB repair pathways, and we do not state otherwise. We have edited the text accordingly to prevent misunderstandings.

Nonetheless, we believe that we have proven that SIRT6 is a sensor, just as MRE11 and KU80 are. We used several lines of experiments (Laser induced damage, biochemical assays and tethering) in addition to published data that allow us to support our final conclusion.

1) SIRT6 arrives to sites of DSBs in less than 5 seconds^10^.

2) SIRT6 arrives to sites of DSBs independently of the known sensors- MRE11 and Ku80.

3) Unlike other upstream factors, SIRT6 arrives to sites of DSBs independently of PARP activity.

4) SIRT6 deficiency results in defective recruitment of a DDR initiator- MRE11.

5) SIRT6 is not recruited by DDR signaling alone- unlike most downstream factors, ATM-LacR signaling does not recruit it.

6) SIRT6 binds ssDNA and sticky-ended DNA with no intermediates in-vitro.

7) Like other DDR initiators, SIRT6 can initiate the DNA damage response even in the absence of actual damage (shown by using SIRT6-LacR at the teething assay).

8) SIRT6 can recruit DDR proteins to the LacO array.

9) SIRT6 deficiency results in defective DDR protein recruitment, demonstrated by the defective recruitment of both 53BP1 and BRCA1.

10) SIRT6 deficiency results in defective repair by both HR and NHEJ^10^.

11) SIRT6 silencing prevents recruitment of BRCA1, 53BP1, RPA, H2Bub120^10^.

When what we expect of a sensor is to: first, recognize and bind DSBs by itself in a fast manner; second, activate DDR protein signaling and recruitment; and last, affect the actual DNA repair, we believe that – when taking all of these findings into consideration – SIRT6 is a DSB sensor.

5) Figure 3A-C. the authors use EMSA to calculate the Kd value and cooperativity for SIRT6-DNA binding. This method is semi quantitative and therefore additional quantitative methods should be used such as Surface Plasmon Resonance (SPR) and Biolayer interferometry (BLI). Moreover, given the fast interaction of SIRT6 with the damaged site (5 seconds), a more reliable Kd value based on the methods listed above can provide additional information for binding kinetics (k_on_ and K_off_) which is needed in order to justify their claims.

Although in theory SPR can provide the kinetic parameters of binding (K_on_, K_off_), it is not a good method for measuring DNA-protein binding due to its generally high binding affinity and strong electrostatic nature. These features give rise to several practical complications:

1) Mass transfer limits on kinetics, where the rates of transfer of components from the injected solution to the immobilized component are slower than the association reaction.

2) Very slow dissociation rates with potential rebinding during the dissociation phase.

3) Limited time for the association reaction due to volume limitations in the injection syringe.

Biolayer interferometry is a relatively new technique and is not widely used. Therefore, we do not know how suitable it is for DNA-binding interactions.

Gel shift assay is a commonly used and highly acceptable method for the evaluation of nucleic acid-protein interactions. One great advantage of this method is that it is not sensitive for binding, meaning that only tightly bound DNA-protein complexes are observable. This is a robust method for Kd that is around a μM range. In our case, it is quite reliable (more reliable than Plasmon resonance.) We believe that more accurate data does not provide additional insight to our findings; these methods are not simple and are beyond the scope of this paper.

6) Several SIRT6 ChIP seq experiments were published, which involves sonicating DNA and the creation of DNA breaks. If SIRT6 binds to ssDNA as the authors propose, it is surprising that ChIP-seq studies were able to identify clear peaks in vivo. This suggests that the ssDNA binding may not occur physiologically in vivo.

When performing ChIP-seq, cells are fixed with formaldehyde prior to their sonication. Thus, DSB sensors as well as other proteins are fixed to their cellular location and are not recruited to the breaks generated through the sonication process. However, if some of them would still bind, DNA damage would occur in the cell randomly (as would occur by sonication as well), thus we do not expect to see an enriched peak, but possibly more “random noise”.

7) For reasons outlined in the rebuttal letter, prior data in Figure 1 is no longer included. The new data are less compelling. Overall effects are very moderate in size. Also, in Figure 1B, it is not clear what n = 10 refers to. Is it cells or perhaps independent experiments? Error bars are not included, and p-values are not shown for Figure 1 data. This is of concern and needs to be addressed. We noted that p-values are stated for other data (but not Figure 1 data) in the Transparent Reporting Form.

The only experiment that was removed from Figure 1 was the downstream factor RNF8. Moreover, MRE11 and KU80 silencing were added, both more relevant since they are the actual known DSB sensors. We fail to understand how that missing panel (RNF8) makes the data less compelling.

In single cell microscopy experiments, since every cell is an individual experiment, n=total number of cells. To our knowledge, stating p-values in this type of experiment is not customary; however, since it was pointed out as a concern, we increased the number of cells in various experiments, adding error bars to all the figures. To analyze the differences between the different graphs we measured the area under the curve. Our results show that while most measurements are characterized by a single time scale, they might differ in their saturation values. Since this makes it impossible to average over the entire data collection, we used an alternative method of analysis based on calculating the area under the graph representing the recruitment vs. time. Physically, this area represents the integrated quantity of recruited protein. These areas, approximated by summing the column of signals, were then analyzed using Prism to extract their p-values. Please see the new figures with SEM and the area under the curve here (Figure 1 and Figure 1—figure supplement 1A). The relevant information was added to the figure legends and the “Transparent reporting form” as well.

8) The authors state in their rebuttal that the U2OS cells used were "very unstable" and that "too many gH2AX foci were present making the experiments impossible to quantify or rely on. When cells were too old or damaged, we thawed new ones from old passages and began the whole experiment again". This seems quite arbitrary and made us concerned about the overall validity and rigor of the studies. At a minimum, more specific criteria for what constitutes cells being "too old or damaged" would need to be clearly described in the manuscript.

As with many cell lines, several passages cause basal levels of DNA damage, increasing the background foci of ɤH2AX. When using the U2OS cells of the tethering assay, if the majority of cells were too “Old” (a term we used before), meaning the cell presented DNA damage without any treatment (exhibited more than 15 ɤH2AX foci per nucleus), cells were discarded and a new vial from an earlier passage was thawed. See example in Author response image 5:

**Author response image 5. respfig5:** IF for ɤH2AX in in SIRT6-LacR-GFP expressing U2OS cells. “New” vs. “Old” cells.

Additional notes:

To better understand whether Lack of proper binding of the HY mutant would affect its recruitment to DNA, we added the lack of recruitment of SIRT6-HY into the laser induced damage. We used SIRT6-KO cells and measured SIRT6-WT or SIRT6-HY to the laser induced damage by live imaging. This experiment proved that SIRT6-HY which is catalytically inactive and has lost its binding capacity by 50% is unable to arrive to the sites of damage (Figure 5C).

[Editors' note: further revisions were suggested prior to acceptance, as described below.]

The manuscript has been improved but there are some remaining issues that need to be addressed before acceptance, as outlined below:1) The revised manuscript now describes more accurately that Parp activity (versus Parp1 protein) is dispensable for SIRT6 binding to DSBs; however, it is problematic that the new submission still fails to show that Parp1 protein does not recruit SIRT6 to breaks. […] If they wish to propose the model that SIRT6 is independent of all known sensors, they will need to state explicitly that this is a speculation, and that their data cannot rule out that recruitment of SIRT6 to DSBs might occur at least in part via its previously described interaction with Parp1.

1) Although we tried, the CRISPER-KO did not work, and it could take time and troubleshooting, therefore we referred to Mao et al. In the paper (in which this recruitment is not the main focus) they present data which shows that PAPR1-KO in MEFs does not affect SIRT6-GFP recruitment to sites of double strand breaks.

However, as we did not do the experiment in our cell lines, and PARP2 and PARP3 could still have influence we agree that the statement should only refer to PARP activity. We revised the test accordingly:

Previous submission: Results:

"Taken together, these results indicate that SIRT6 arrives to the sites of damage independently of other known DSB sensors, and that in the absence of actual DNA damage, signaling itself is not sufficient to bring SIRT6 to the damage sites."

Revised submission:

"… SIRT6 arrives to the sites of damage independently of MRE11, Ku80 and PARP activity, and that in the absence…"

Previous submission: Discussion:

"In addition, we showed that SIRT6 can arrive at the sites of DSBs independently of the known sensors, and activate the DDR on its own."

Revised submission:

"In addition, we showed that SIRT6 can arrive at the sites of DSBs independently of the known sensors MRE11 and Ku80 as well as PARP activity, and activate the DDR on its own."

2) There is also remaining concern that conclusions continue to be overstated in places. In the last sentence of the Discussion ("In conclusion, we have demonstrated that Sirtuins – and mainly SIRT6 – have a role as independent DNA damage sensors."), the conclusion regarding all Sirtuins is not supported by the data. The authors should limit their conclusion to SIRT6. There is not sufficient evidence from their experiments that the other Sirtuins have roles as sensors.

We agree with this comment and the new phrase in the Discussion states:

"In conclusion, we have demonstrated that SIRT6 has a role as an independent DNA damage sensor"

**References**

1. Mao, Z. et al. SIRT6 promotes DNA repair under stress by activating PARP1. *Science***332,** 1443–6 (2011).

2. Haince, J.-F. F. et al. PARP1-dependent kinetics of recruitment of MRE11 and NBS1 proteins to multiple DNA damage sites. *J. Biol. Chem.***283,** 1197–208 (2008).

3. Soutoglou, E. and Misteli, T. Activation of the cellular DNA damage response in the absence of DNA lesions. *Science***320,** 1507–10 (2008).

4. Batenburg, N. L. et al. CSB interacts with BRCA1 in late S/G2 to promote MRN- and CtIP-mediated DNA end resection. *Nucleic Acids Res.***47,** 10678–10692 (2019).

5. Roukos, V., Burgess, R. C. and Misteli, T. Generation of cell-based systems to visualize chromosome damage and translocations in living cells. *Nat Protoc***9,** 2476–92 (2014).

6. Helfricht, A. et al. Remodeling and spacing factor 1 (RSF1) deposits centromere proteins at DNA double-strand breaks to promote non-homologous end-joining. *Cell Cycle***12,** 3070–82 (2013).

7. Luijsterburg, M. S. et al. A new non-catalytic role for ubiquitin ligase RNF8 in unfolding higher-order chromatin structure. *EMBO J.***31,** 2511–27 (2012).

8. Zolghadr, K. et al. A fluorescent two-hybrid assay for direct visualization of protein interactions in living cells. *Mol. Cell Proteomics***7,** 2279–87 (2008).

9. Jacome, A. and Fernandez-Capetillo, O. Lac operator repeats generate a traceable fragile site in mammalian cells. *EMBO Rep.***12,** 1032–8 (2011).

10. Toiber, D. et al. SIRT6 recruits SNF2H to DNA break sites, preventing genomic instability through chromatin remodeling. *Mol. Cell***51,** 454–68 (2013).